# Categorical Normalizing Flows via Continuous Transformations

**Phillip Lippe**
University of Amsterdam, QUVA lab
p.lippe@uva.nl

**Efstratios Gavves**
University of Amsterdam
egavves@uva.nl

## Abstract

Despite their popularity, to date, the application of normalizing flows on categorical data stays limited. The current practice of using dequantization to map discrete data to a continuous space is inapplicable as categorical data has no intrinsic order. Instead, categorical data have complex and latent relations that must be inferred, like the synonymy between words. In this paper, we investigate *Categorical Normalizing Flows*, that is normalizing flows for categorical data. By casting the encoding of categorical data in continuous space as a variational inference problem, we jointly optimize the continuous representation and the model likelihood. Using a factorized decoder, we introduce an inductive bias to model any interactions in the normalizing flow. As a consequence, we do not only simplify the optimization compared to having a joint decoder, but also make it possible to scale up to a large number of categories that is currently impossible with discrete normalizing flows. Based on Categorical Normalizing Flows, we propose GraphCNF a permutation-invariant generative model on graphs. GraphCNF implements a three-step approach modeling the nodes, edges, and adjacency matrix stepwise to increase efficiency. On molecule generation, GraphCNF outperforms both one-shot and autoregressive flow-based state-of-the-art.

## 1 Introduction

Normalizing Flows have been popular for tasks with continuous data like image modeling (Dinh et al., 2017; Kingma and Dhariwal, 2018; Ho et al., 2019) and speech generation (Kim et al., 2019; Prenger et al., 2019) by providing efficient parallel sampling and exact density evaluation. The concept that normalizing flows rely on is the rule of change of variables, a continuous transformation designed for continuous data. However, there exist many data types typically encoded as discrete, categorical variables, like language and graphs, where normalizing flows are not straightforward to apply.

To address this, it has recently been proposed to discretize the transformations inside normalizing flows to act directly on discrete data. Unfortunately, these discrete transformations have shown to be limited in terms of the vocabulary size and layer depth due to gradient approximations (Hoogeboom et al., 2019; Tran et al., 2019). For the specific case of discrete but ordinal data, like images where integers represent quantized values, a popular strategy is to add a small amount of noise to each value (Dinh et al., 2017; Ho et al., 2019). It is unnatural, however, to apply such dequantization techniques for the general case of categorical data, where values represent categories with no intrinsic order. Treating these categories as integers for dequantization biases the data to a non-existing order, and makes the modeling task significantly harder. Besides, relations between categories are often multi-dimensional, for example, word meanings, which cannot be represented with dequantization.

In this paper, we investigate normalizing flows for the general case of categorical data. To account for discontinuity, we propose continuous encodings in which different categories correspond to unique, non-overlapping and thus close-to-deterministic volumes in a continuous latent space. Instead of pre-specifying the non-overlapping volumes per category, we resort to variational inference to jointly learn those and model the likelihood by a normalizing flow at the same time. This work is not the first to propose variational inference with normalizing flows, mostly considered for improving the flexibility of the approximate posterior (Kingma et al., 2016; Rezende and Mohamed, 2015; Van Den Berg et al., 2018). Different from previous works, we use variational inference to learn

a continuous representation $z$ of the discrete categorical data $x$ to a normalizing flow. A similar idea has been investigated in (Ziegler and Rush, 2019), who use a variational autoencoder structure with the normalizing flow being the prior. As both their decoder and normalizing flow model (complex) dependencies between categorical variables, (Ziegler and Rush, 2019) rely on intricate yet sensitive learning schedules for balancing the likelihood terms. Instead, we propose to separate the representation and relation modeling by factorizing the decoder both over the categorical variable $x$ and the conditioning latent $z$. This forces the encoder and decoder to focus only on the mapping from categorical data to continuous encodings, and not model any interactions. By inserting this inductive bias, we move all complexity into the flow. We call this approach *Categorical Normalizing Flows* (CNF).

Categorical Normalizing Flows can be applied to any task involving categorical variables, but we primarily focus on modeling graphs. Current state-of-the-art approaches often rely on autoregressive models (Li et al., 2018; Shi et al., 2020; You et al., 2018) that view graphs as sequences, although there exists no intrinsic order of the node. In contrast, normalizing flows can perform generation in parallel making a definition of order unnecessary. By treating both nodes and edges as categorical variables, we employ our variational inference encoding and propose GraphCNF. GraphCNF is a novel permutation-invariant normalizing flow on graph generation which assigns equal likelihood to any ordering of nodes. Meanwhile, GraphCNF efficiently encodes the node attributes, edge attributes, and graph structure in three consecutive steps. As shown in the experiments, the improved encoding and flow architecture allows GraphCNF to outperform significantly both the autoregressive and parallel flow-based state-of-the-art. Further, we show that Categorical Normalizing Flows can be used in problems with regular categorical variables like modeling natural language or sets.

Our contributions are summarized as follows. Firstly, we propose Categorical Normalizing Flows using variational inference with a factorized decoder to move all complexity into the prior and scale up to large number of categories. Secondly, starting from the Categorical Normalizing Flows, we propose GraphCNF, a permutation-invariant normalizing flow on graph generation. On molecule generation, GraphCNF sets a new state-of-the-art for flow-based methods outperforming one-shot and autoregressive baselines. Finally, we show that simple mixture models for encoding distributions are accurate, efficient, and generalize across a multitude of setups, including sets language and graphs.

## 2 CATEGORICAL NORMALIZING FLOWS

### 2.1 NORMALIZING FLOWS ON CONTINUOUS DATA

A normalizing flow (Rezende and Mohamed, 2015; Tabak and Vanden Eijnden, 2010) is a generative model that models a probability distribution $p(\boldsymbol{z}^{(0)})$ by applying a sequence of invertible, smooth mappings $f_1, ..., f_K : \mathbb{R}^d \to \mathbb{R}^d$. Using the rule of change of variables, the likelihood of the input $\boldsymbol{z}^{(0)}$ is determined as follows:

$$p(\boldsymbol{z}^{(0)}) = p(\boldsymbol{z}^{(K)}) \cdot \prod_{k=1}^{K} \left| \det \frac{\partial f_k(\boldsymbol{z}^{(k-1)})}{\partial \boldsymbol{z}^{(k-1)}} \right| \tag{1}$$

where $\boldsymbol{z}^{(k)} = f_k(\boldsymbol{z}^{(k-1)})$, and $p(\boldsymbol{z}^{(K)})$ represents a prior distribution. This calculation requires to compute the Jacobian for the mappings $f_1, ..., f_K$, which is expensive for arbitrary functions. Thus, the mappings are often designed to allow efficient computation of its determinant. One of such is the coupling layer proposed by Dinh et al. (2017) which showed to work well with neural networks. For a detailed introduction to normalizing flows, we refer the reader to Kobyzev et al. (2019).

### 2.2 NORMALIZING FLOWS ON CATEGORICAL DATA

We define $\boldsymbol{x} = \{x_1, ..., x_S\}$ to be a multivariate, categorical random variable, where each element $x_i$ is itself a categorical variable of $K$ categories with no intrinsic order. For instance, $\boldsymbol{x}$ could be a sentence with $x_i$ being the words. Our goal is to learn the joint probability mass function, $P_{\text{model}}(\boldsymbol{x})$, via a normalizing flow. Specifically, as normalizing flows constitute a class of continuous transformations, we aim to learn a continuous latent space in which each categorical choice of a variable $x_i$ maps to a stochastic continuous variable $\boldsymbol{z}_i \in \mathbb{R}^d$ whose distribution we learn.

Compared to variational autoencoders (Kingma and Welling, 2014) and latent normalizing flows (Ziegler and Rush, 2019), we want to ensure that all modeling complexity is solely in the prior, and keep a lossless reconstruction from latent space. To implement this, we simplify the decoder by factorizing the decoder over latent variables: $p(\boldsymbol{x}|\boldsymbol{z}) = \prod_i p(x_i|\boldsymbol{z}_i)$. Factorizing the conditional likelihood means that we enforce independence between the categorical variables $x_i$ given their learned continuous encodings $\boldsymbol{z}_i$. Therefore, any interaction between the categorical variables $\boldsymbol{x}$ must be learned inside the normalizing flow. If in this setup, the encoding distributions of multiple categories would overlap, the prior would be limited in the dependencies over $x_1, ..., x_S$ it can model as it cannot clearly distinguish between all categories. Therefore, the encoder $q(\boldsymbol{z}|\boldsymbol{x})$ is being optimized to provide suitable representations of the categorical variables to the flow while separating the different categories in latent space. Meanwhile, the decoder is incentivized to be deterministic, i.e. precisely reconstructing $x$ from $z$, in order to minimize the overlap of categories. Overall, our objective becomes:

$$\mathbb{E}_{\boldsymbol{x} \sim P_{\text{data}}}[\log P_{\text{model}}(\boldsymbol{x})] \geq \mathbb{E}_{\boldsymbol{x} \sim P_{\text{data}}}\mathbb{E}_{\boldsymbol{z} \sim q(\cdot|\boldsymbol{x})}\left[\log \frac{p_{\text{model}}(\boldsymbol{z})\prod_i p(x_i|\boldsymbol{z}_i)}{q(\boldsymbol{z}|\boldsymbol{x})}\right] \qquad (2)$$

We refer to this framework as *Categorical Normalizing Flows*. In contrast to dequantization, the continuous encoding $\boldsymbol{z}$ is not bounded by the domain of the encoding distribution. Instead, the partitioning is jointly learned with the model likelihood. Furthermore, we can freely choose the dimensionality of the continuous variables, $\boldsymbol{z}_i$, to fit the number of categories and their relations.

**Modeling the encoder** The encoder $q(\boldsymbol{z}|\boldsymbol{x})$ and decoder $p(x_i|\boldsymbol{z}_i)$ can be implemented in several ways. The first and main setup we consider is to encode each category by a logistic distribution with a learned mean and scaling. Therefore, our encoding distribution $q(\boldsymbol{z}_i)$ is a mixture of $K$ logistics, one per category. With $g$ denoting the logistic, the encoder becomes $q(\boldsymbol{z}|\boldsymbol{x}) = \prod_{i=1}^{S} g(\boldsymbol{z}_i|\mu(x_i), \sigma(x_i))$. In this setup, the decoder likelihood can actually be found correspondingly to the encoder by applying Bayes: $p(x_i|\boldsymbol{z}_i) = \frac{\tilde{p}(x_i)q(\boldsymbol{z}_i|x_i)}{\sum_{\hat{x}} \tilde{p}(\hat{x})q(\boldsymbol{z}_i|\hat{x})}$ with $\tilde{p}(x_i)$ being a prior over categories. Hence, we do not need to learn a separate decoder but can calculate the likelihood based on the encoder's parameters. The objective in Equation 2 simplifies to the following:

$$\mathbb{E}_{\boldsymbol{x} \sim P_{\text{data}}}[\log P_{\text{model}}(\boldsymbol{x})] \geq \mathbb{E}_{\boldsymbol{x} \sim P_{\text{data}}}\mathbb{E}_{\boldsymbol{z} \sim q(\cdot|\boldsymbol{x})}\left[\log\left(p_{\text{model}}(\boldsymbol{z})\prod_{i=1}^{S} \frac{\tilde{p}(x_i)}{\sum_{\hat{x}} \tilde{p}(\hat{x})q(\boldsymbol{z}_i|\hat{x})}\right)\right] \qquad (3)$$

Note that the term $q(\boldsymbol{z}_i|x_i)$ in the numerator of $p(x_i|\boldsymbol{z}_i)$ cancels out with the denominator in Equation 2. Given that the encoder and decoder are sharing the parameters, we remove any possible mismatch between $p(x_i|\boldsymbol{z}_i)$ and $q(x_i|\boldsymbol{z}_i)$. This allows changes in the encoding distribution to directly being propagated to the decoder, and further moves the focus of the training to the prior. Besides, the mixture encoding introduces a dependency of the true posterior $p(\boldsymbol{z}|\boldsymbol{x})$ on the approximate posterior $q(\boldsymbol{z}|\boldsymbol{x})$, which potentially tightens the variational gap compared to a separately learned decoder. During testing, we can use importance sampling (Burda et al., 2016) to further reduce the gap. Details on the posterior dependency in the variational gap, and training and test steps can be found in Appendix A.1.

The mixture model is simple and efficient, but might be limited in the distributions it can express. To test whether greater encoding flexibility is needed, we experiment with adding flows conditioned on the categories which transform each logistic into a more complex distribution. We refer to this approach as linear flows. Taking a step further, we can also represent the encoder $q(\boldsymbol{z}|\boldsymbol{x})$ with a flow across categorical variables, similar to variational dequantization (Ho et al., 2019). Experiments presented in Section 5 show however that a simple mixture of logistics usually suffices.

## 3 GRAPH GENERATION WITH CATEGORICAL NORMALIZING FLOWS

A graph $\mathcal{G} = (V, E)$ is defined by a set of nodes $V$, and a set of edges $E$ representing connections between nodes. When modeling a graph, we must take into account the node and edge attributes, often represented by categorical data, as well as the overall graph structure. Moreover, nodes and edges are better viewed as sets and not sequences since any permutation represents the same graph and assigned the same likelihood.

We propose GraphCNF, a normalizing flow for graph generation, which is invariant to the order of nodes by generating all nodes and edges at once. Given a graph $\mathcal{G}$, we model each node and edge as a

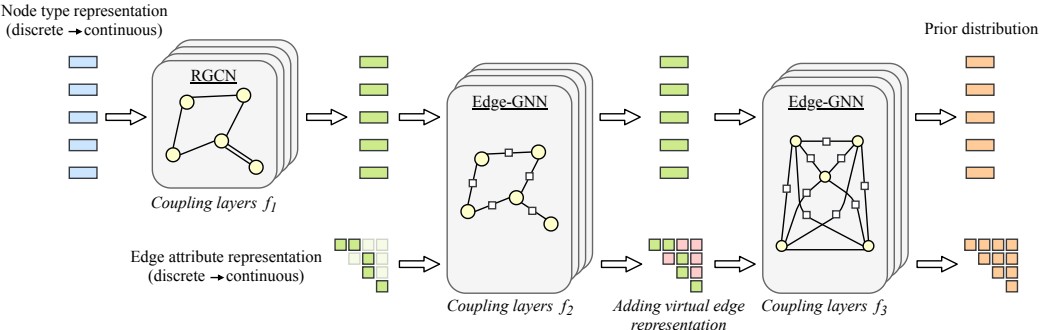

Figure 1: Visualization of GraphCNF for an example graph of five nodes. We add the node and edge attributes, as well as the virtual edges stepwise to the latent space while leveraging the graph structure in the coupling layers. The last step considers a fully connected graph with features per edge.

separate categorical variable where the categories correspond to their discrete attributes. To represent the graph structure, i.e. between which pairs of nodes does or does not exist an edge, we add an extra category to the edges representing the missing or *virtual* edges. Hence, to model an arbitrary graph, we consider an edge variable for every possible tuple of nodes.

To apply normalizing flows on the node and edge categorical variables, we map them into continuous latent space using Categorical Normalizing Flows. Subsequent coupling layers map those representations to a continuous prior distribution. Thereby, GraphCNF uses two crucial design choices for graph modeling: (1) we perform the generation stepwise by first encoding the nodes, edges and then the adjacency matrix for improved efficiency, and (2) we introduce an inductive bias that the model assigns equal likelihood to any ordering of the nodes.

### 3.1 THREE-STEP GENERATION

Modeling all edges including the virtual ones requires a significant amount of latent variables and is computationally expensive. However, normalizing flows have been shown to benefit from splitting of latent variables at earlier layers while increasing efficiency (Dinh et al., 2017; Kingma and Dhariwal, 2018). Thus, we propose to add the node types, edge attributes and graph structure stepwise to the latent space as visualized in Figure 1. In the first step, we encode the nodes into continuous latent space, $z_0^{(V)}$, using Categorical Normalizing Flows. On those, we apply a group of coupling layers, $f_1$, which additionally use the adjacency matrix and the edge attributes, denoted by $E_{attr}$, as input. Thus, we can summarize the first step as:

$$z_1^{(V)} \quad = \quad f_1\big(z_0^{(V)}; E, E_{attr}\big) \tag{4}$$

The second step incorporates the edge attributes, $E_{attr}$, into latent space. Hence, all edges of the graph except the virtual edges are encoded into latent variables, $z_0^{(E_{attr})}$, representing their attribute. The following coupling layers, denoted by $f_2$, transform both the node and edge attribute variables:

$$z_2^{(V)}, z_1^{(E_{attr})} \quad = \quad f_2\big(z_1^{(V)}, z_0^{(E_{attr})}; E\big) \tag{5}$$

Finally, we add the virtual edges to the latent variable model as $z_0^{(E^*)}$. Thereby, we need to slightly adjust our encoding from Categorical Normalizing Flows as we consider the virtual edges as an additional category of the edges. While the other categories are already encoded by $z_1^{(E_{attr})}$, we add a separate encoding distribution for the virtual edges, for which we use a simple logistic. Meanwhile, the decoder needs to be applied on all edges, as we need to distinguish the continuous representation between virtual and non-virtual edges. Overall, the mapping can be summarized as:

$$z_3^{(V)}, z_1^{(E)} \quad = \quad f_3\big(z_2^{(V)}, z_0^{(E)}\big) \quad \text{where} \quad z_0^{(E)} = \big[z_1^{(E_{attr})}, z_0^{(E^*)}\big] \tag{6}$$

where the latent variables $z_3^{(V)}$ and $z_1^{(E)}$ are trained to follow a prior distribution. During sampling, we first inverse $f_3$ and determine the general graph structure. Next, we inverse $f_2$ and reconstruct the edge attributes. Finally, we apply the inverse of $f_1$ and determine the node types.

## 3.2 PERMUTATION-INVARIANT GRAPH MODELING

To achieve permutation invariance for the likelihood estimate, the transformations of the coupling layers need to be independent of the node order. This includes both the split of variables that will be transformed and the network model that predicts the transformation parameters. We ensure the first aspect by applying a channel masking strategy (Dinh et al., 2017), where the split is performed over the latent dimensions for each node and edge separately making it independent of the node order. For the second aspect, we leverage the graph structure in the coupling networks and apply graph neural networks. In the first step of GraphCNF, $f_1$, we use a Relation GCN (Schlichtkrull et al., 2018) which incorporates the categorical edge attributes into the layer. For the second and third steps, we need a graph network that supports the modeling of both node and edge features. We implement this by alternating between updates of the edge and the node features. Specifically, given node features $\boldsymbol{v}^t$ and edge features $\boldsymbol{e}^t$ at layer $t$, we update those as follows:

$$\boldsymbol{v}^{t+1} = f_{node}(\boldsymbol{v}^t; \boldsymbol{e}^t), \qquad \boldsymbol{e}^{t+1} = f_{edge}(\boldsymbol{e}^t; \boldsymbol{v}^{t+1}) \tag{7}$$

We call this network Edge-GNN, and compare different implementations of $f_{node}$ and $f_{edge}$ in Appendix B. Using both design choices, GraphCNF models a permutation invariant distribution.

## 4 RELATED WORK

**Dequantization** Applying continuous normalizing flows on discrete data leads to undesired density models where arbitrarily high likelihoods are placed on particular values (Theis et al., 2016; Uria et al., 2013). A common solution to this problem is to *dequantize* the data $\boldsymbol{x}$ by adding noise $\boldsymbol{u} \in [0, 1)^D$. Theis et al. (2016) have shown that modeling $p_{\text{model}}(\boldsymbol{x} + \boldsymbol{u})$ lower-bounds the discrete distribution $P_{\text{model}}(\boldsymbol{x})$. The noise distribution $q(\boldsymbol{u}|\boldsymbol{x})$ is usually uniform or learned by a second normalizing flow. The latter is referred to as *variational dequantization* and has been proven to be crucial for state-of-the-art image modeling (Ho et al., 2019; Hoogeboom et al., 2020). Categories, however, are not quantized values, so that ordering them as integers introduces bias to the representation.

**Discrete NF** Recent works have investigated normalizing flows with discretized transformations. Hoogeboom et al. (2019) proposed to use additive coupling layers with rounding operators for ensuring discrete output. Tran et al. (2019) discretizes the output by a Gumbel-Softmax approximating an argmax operator. Thereby, the coupling layers resemble a reversible shift operator. While both approaches achieved competitive results, due to discrete operations the gradient approximations have been shown to introduce new challenges, such as limiting the number of layers or distribution size.

**Latent NF** Several works have investigated the application of normalizing flows in variational auto-encoders (VAEs) (Kingma and Welling, 2014) for increasing the flexibility of the approximate posterior (Kingma et al., 2016; Van Den Berg et al., 2018; Tomczak and Welling, 2017). However, VAEs model a lower bound of the true likelihood. To minimize this gap, Ziegler and Rush (2019) proposed Latent Normalizing Flows that move the main model complexity into the prior by using normalizing flows. In contrast to CNFs, Latent NF have a joint decoder over the latent space, $p(\boldsymbol{x}|\boldsymbol{z})$, which allows the modeling of interactions between variables in the decoder. Thus, instead of an inductive bias to push all complexity to the normalizing flow, Latent NF rely on a loss-scheduling weighting the decoder loss much higher. This pushes the decoder to be deterministic but can lead to unstable optimization due to neglecting the flow's likelihood. Further, experiments on sequence tasks show Latent NF to be competitive but are still outperformed by an LSTM baseline as we observed in our experiments.

**Graph modeling** The first generation models on graphs have been autoregressive (Liao et al., 2019; You et al., 2018), generating nodes and edges in sequential order. While being efficient in memory, they are slow in sampling and assume an order in the set of nodes. The first application of normalizing flows for graph generation was introduced by Liu et al. (2019), where a flow modeled the node representations of a pretrained autoencoder. Recent works of GraphNVP (Madhawa et al., 2019) and GraphAF (Shi et al., 2020) proposed normalizing flows for molecule generation. GraphNVP consists of two separate flows, one for modeling the adjacency matrix and a second for modeling the node types. Although allowing parallel generation, the model is sensitive to the node order due to its masking strategy and feature networks in the coupling layer. GraphAF is an autoregressive normalizing flow sampling nodes and edges sequentially but allowing parallel training. However, both flows use standard uniform dequantization to represent the node and edge categories. VAE have

also been proposed for latent-based graph generation (Simonovsky and Komodakis, 2018; Ma et al., 2018; Liu et al., 2018; Jin et al., 2018). Although those models can be permutation-invariant, they model a lower bound and do not provide a lossless reconstruction from latent space.

# 5 EXPERIMENTAL RESULTS

We start our experiments by evaluating GraphCNF on two benchmarks for graph generation, namely molecule generation and graph coloring. Further, to test generality we evaluate CNFs on other categorical problems, specifically language and set modeling. For the normalizing flows, we use a sequence of logistic mixture coupling layers (Ho et al., 2019) mapping a mixture of logistic distributions back into a single mode. Before each coupling layer, we include an activation normalization layer and invertible 1x1 convolution (Kingma and Dhariwal, 2018). For reproducibility, we provide all hyperparameter details in Appendix D, and make our code publicly available.[1]

## 5.1 MOLECULE GENERATION

Modeling and generating graphs is crucial in biology and chemistry for applications such as drug discovery, where molecule generation has emerged as a common benchmark (Jin et al., 2018; Shi et al., 2020). In a molecule graph, the nodes are atoms and the edges represent bonds between atoms, both represented by categorical features. Using a dataset of existing molecules, the goal is to learn a distribution of valid molecules as not all possible combinations of atoms and bonds are valid. We perform experiments on the Zinc250k (Irwin et al., 2012) dataset which consists of 250,000 drug-like molecules. The molecules contain up to 38 atoms of 9 different types, with three different bond types possible between the atoms. For comparability, we follow the preprocessing of Shi et al. (2020).

We compare GraphCNF to baselines that consider molecules as a graph and not as text representation. As per VAE-based approaches, we consider R-VAE (Ma et al., 2018) and Junction-Tree VAE (JT-VAE) (Jin et al., 2018). R-VAE is a one-shot generation model using regularization to ensure semantic validity. JT-VAE represents a molecule as a junction tree of sub-graphs that are obtained from the training dataset. We also compare our model to GraphNVP (Madhawa et al., 2019) and GraphAF (Shi et al., 2020). The models are evaluated by sampling 10,000 examples and measuring the proportion of valid molecules. We also report the proportion of unique molecules and novel samples that are not in the training dataset. These metrics prevent models from memorizing a small subset of graphs. Finally, the reconstruction rate describes whether graphs can be accurately decoded from latent space. Normalizing Flows naturally score 100% due to their invertible mapping, and we achieve the same with our encoding despite no guarantees.

Table 1 shows that GraphCNF generates almost twice as many valid molecules than other one-shot approaches. Yet, the validity and uniqueness stay at almost 100%. Even the autoregressive normalizing flow, GraphAF, is outperformed by GraphCNF by 15%. However, the rules for generating valid molecules can be enforced in autoregressive models by masking out the invalid outputs. This has been the case for JT-VAE as it has been trained with those manual rules, and thus achieves a

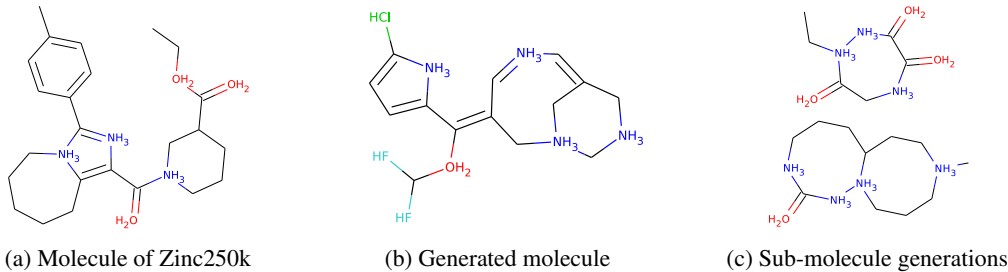

(a) Molecule of Zinc250k      (b) Generated molecule      (c) Sub-molecule generations

Figure 2: Visualization of molecules from (a) the Zinc250k dataset and (b,c) generated by GraphCNF. Nodes with black connections and no description represent carbon atoms. Sub-figure (c) shows the failure case of two valid sub-graphs. More example generations can be found in Appendix C.

---

[1]Code available here: `https://github.com/phlippe/CategoricalNF`

Table 1: Performance on molecule generation trained on Zinc250k (Irwin et al., 2012), calculated on 10k samples and averaged over 4 runs. The standard deviation of those runs can be found in Appendix C. Scores of the baselines are taken from their respective papers.

| Method | Validity | Uniqueness | Novelty | Reconstruction | Parallel | Manual Rules |
|--------|----------|-----------|---------|----------------|----------|--------------|
| JT-VAE [10] | 100% | 100% | 100% | 71% | ✗ | ✓ |
| GraphAF [36] | 68% | 99.10% | 100% | 100% | ✗ | ✗ |
| R-VAE [23] | 34.9% | 100% | – | 54.7% | ✓ | ✗ |
| GraphNVP [24] | 42.60% | 94.80% | 100% | 100% | ✓ | ✗ |
| GraphCNF | 83.41% | 99.99% | 100% | 100% | ✓ | ✗ |
| + Sub-graphs | 96.35% | 99.98% | 99.98% | 100% | ✓ | ✗ |

validity of 100%. Nevertheless, we are mainly interested in the model's capability of learning the rules by itself and being not specific to any application. While GraphNVP and GraphAF sample with a lower standard deviation from the prior to increase validity, we explicitly sample from the original prior to underline that our model covers the whole latent space well. Surprisingly, we found out that most invalid graphs consist of two or more that in isolation are valid, as shown in Figure 2c. This can happen as one-shot models have no guidance for generating a single connected graph. By taking the largest sub-graph of these predictions, we obtain a validity ratio of 96.35% making our model generate almost solely valid molecules *without any manually encoded rules*. We also evaluated our model on the Moses (Polykovskiy et al., 2018) dataset and achieved similar scores as shown in Appendix C.

## 5.2 GRAPH COLORING

Graph coloring is a well-known combinatorial problem (Bondy et al., 1976) where for a graph $\mathcal{G}$, each node is assigned to one of $K$ colors. Yet, two adjacent nodes cannot have the same color (see Figure 3). Modeling the distribution of valid color assignments to arbitrary graphs is NP-complete. To train models on such a distribution, we generate a dataset of valid graph colorings for randomly sampled graphs. To further investigate the effect of complexity, we create two dataset versions, one with graphs of size $10 \leq |V| \leq 20$ and another with $25 \leq |V| \leq 50$, as larger graphs are commonly harder to solve.

Figure 3: Example graph with $|V| = 26$ and generated valid coloring by GraphCNF.

For graph coloring, we rely on GraphCNF and compare to a variational autoencoder and an autoregressive model generating one node at a time. As no edges are being modeled here, we only use the first step of GraphCNF's three-step generation. For all models, we apply the same Graph Attention network (Veličković et al., 2018). As autoregressive models require a manually prescribed node order, we compare the following: a *random* ordering per graph, *largest_first* which is inspired by heuristics of automated theorem provers that start from the nodes with the most connections, and *smallest_first*, where we reverse the order of the previous heuristic. We evaluate the models by measuring the likelihood of color assignments to unseen test graphs in bits per node. Secondly, we sample one color assignment per model for each test graph and report the proportion of valid colorings.

The results in Table 2 show that the node ordering has indeed a significant effect on the autoregressive model's performance. While the smallest_first ordering leads to only 32% valid solutions on the large dataset, reversing the order simplifies the task for the model such that it generates more than twice as many valid color assignments. In contrast, GraphCNF is invariant of the order of nodes. Despite generating all nodes in parallel, it outperforms all node orderings on the small dataset, while being close to the best ordering on the larger dataset. This invariance property is especially beneficial in tasks where an optimal order of nodes is not known, like molecule generation. Although having more parameters, the sampling with GraphCNF is also considerably faster than the autoregressive models. The sampling time can further be improved by replacing the logistic mixture coupling layers with

Table 2: Results on the graph coloring problem. Runtimes are measured on a NVIDIA TitanRTX GPU with a batch size of 128. The standard deviations of the results can be found in Appendix D.2.

| Method | $10 \leq |V| \leq 20$ | | | $25 \leq |V| \leq 50$ | | |
| --- | --- | --- | --- | --- | --- | --- |
| | Validity | Bits per node | Time | Validity | Bits per node | Time |
| VAE | 44.95% | 0.84bpd | 0.05s | 7.75% | 0.64bpd | 0.10s |
| RNN+Smallest_first | 76.86% | 0.73bpd | 0.69s | 32.27% | 0.50bpd | 2.88s |
| RNN+Random | 88.62% | 0.70bpd | 0.69s | 49.28% | 0.46bpd | 2.88s |
| RNN+Largest_first | 93.41% | 0.68bpd | 0.69s | **71.32%** | **0.43**bpd | 2.88s |
| GraphCNF | **94.56%** | **0.67**bpd | 0.28s | 66.80% | 0.45bpd | 0.54s |
| $-$ Affine coupling | 93.90% | 0.69bpd | 0.12s | 65.78% | 0.47bpd | 0.35s |

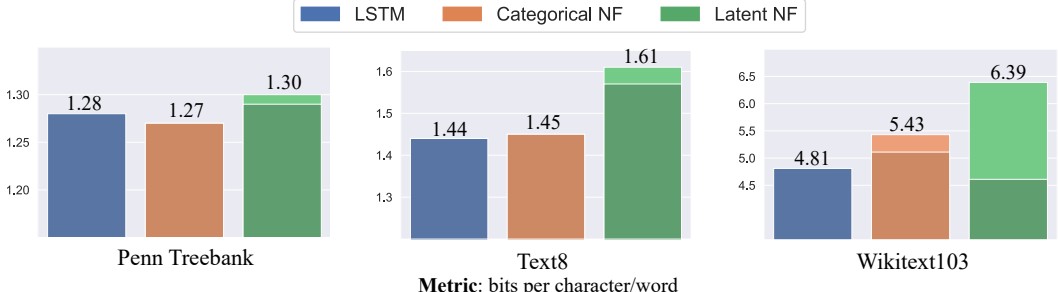

Figure 4: Results on language modeling. The reconstruction error is shown in a lighter color corresponding to the model. The exact results including standard deviations can be found in Table 11.

affine ones. Due to the lower complexity, we see a slight decrease in validity and bits per dimension but can verify that logistic mixture couplings are not crucial for CNFs.

## 5.3 Language modeling

To compare CNF with Latent NF, we test both models on language modeling. We experiment with two popular character-level datasets, Penn Treebank (Marcus et al., 1994) and text8 (Mahoney, 2011) with a vocabulary size of $K = 51$ and $K = 27$ respectively. We also test a word-level dataset, Wikitext103 (Merity et al., 2017), with $K = 10,000$ categories, which Discrete NF cannot handle due to its gradient approximations (Tran et al., 2019). We follow the setup of Ziegler and Rush (2019) for the Penn Treebank and train on sequences of 256 tokens for the other two datasets. Both Latent NF and CNF apply a single mixture coupling layer being autoregressive across time and latent dimensions and differ only in the encoding/decoding strategy. The applied LSTM in the coupling layers is shown as an additional RNN baseline in Figure 4. Categorical Normalizing Flow performs on par with their autoregressive baseline, only slightly underperforming on Wikitext103 due to using a single flow layer. Latent NF however performs considerably worse on text8 and Wikitext103 due to a non-deterministic decoding and higher fluctuation in the loss that we experienced during training (see Appendix A.4.2 for visualizations). This shows the importance of a factorized likelihood and underlines the two benefits of CNFs. Firstly, CNFs are more stable and simpler to train as no loss scheduling is required, and the likelihood is mainly modeled in the flow prior. Secondly, the encoding of CNFs is much more efficient than Latent NFs. We conclude that CNFs are more widely applicable, and Latent NFs might be the better choice if the decoder can explicitly help, *i.e.*, prior knowledge.

## 5.4 Set modeling

Finally, we present experiments on sets of categorical variables, for which we create two toy datasets with known data likelihood: *set shuffling* and *set summation*. The goal is to assign high likelihood only to those possible sets that occur in the dataset, and shows how accurately our flows can model an arbitrary, discrete dataset. In set shuffling, we model a set of $N$ categorical variables each having one out of $N$ categories. Each category has to appear exactly once, which leads to $N!$ possible

Table 3: Results on set modeling. Metric used is bits per categorical variable (dimension).

| Model | Set shuffling | Set summation |
|---|---|---|
| Discrete NF (Tran et al., 2019) | 3.87 ±0.04 | 2.51 ±0.00 |
| Variational Dequant. (Ho et al., 2019) | 3.01 ±0.02 | 2.29 ±0.01 |
| Latent NF (Ziegler and Rush, 2019) | **2.78** ±0.00 | 2.26 ±0.01 |
| CNF + Mixture model | **2.78** ±0.00 | **2.24** ±0.00 |
| CNF + Linear flows | **2.78** ±0.00 | 2.25 ±0.00 |
| CNF + Variational Encoding | 2.79 ±0.01 | 2.25 ±0.01 |
| Optimal | 2.77 | 2.24 |

assignments that need to be modeled. In set summation, we again consider a set of size $N$ with $N$ categories, but those categories represent the actual integers $1, 2, ..., N$ and have to sum to an arbitrary number, $L$. In contrast to set shuffling, the data is ordinal, which we initially expected to help dequantization methods. For both experiments we set $N = 16$ and $L = 42$.

In Table 3, we compare CNFs to applying variational dequantization (Ho et al., 2019), Latent Normalizing Flows (Ziegler and Rush, 2019) and Discrete Normalizing Flows (Tran et al., 2019). The results show that CNFs achieve nearly optimal performance. Although we model a lower bound in continuous space, our flows can indeed model discrete distributions precisely. Interestingly, representing the categories by a simple mixture model is sufficient for achieving these results. We observe the same trend in domains with more complex relations between categories, such as on graphs and language modeling, presumably because of both the coupling layers and the prior distribution resting upon logistic distributions as well. Variational dequantization performs worse on the shuffling dataset, while on set summation with ordinal data, the gap to the optimum is smaller. The same holds for Discrete NFs, although it is worth noting that unlike CNFs, optimizing Discrete NFs had issues due to their gradient approximations. Latent Normalizing Flows with the joint decoder achieve similar performance as CNF, which can be attributed to close-to deterministic decoding. When looking at the encoding space (see Appendix A.4.1 for visualization), we see that Latent NF has indeed learned a mixture model as well. Hence, the added complexity is not needed on this simple dataset.

## 6 CONCLUSION

We present Categorical Normalizing Flows which learn a categorical, discrete distribution by jointly optimizing the representation of categorical data in continuous latent space and the model likelihood of a normalizing flow. Thereby, we apply variational inference with a factorized posterior to maintain almost unique decoding while allowing flexible encoding distributions. We find that a plain mixture model is sufficient for modeling discrete distributions accurately while providing an efficient way for encoding and decoding categorical data. Compared to a joint posterior, CNFs are more stable, efficient, and have an inductive bias to move all modeling complexity into the prior. Furthermore, GraphCNF, a normalizing flow on graph modeling based on CNFs, outperforms autoregressive and one-shot approaches on molecule generation and graph coloring while being invariant to the node order. This emphasizes the potential of normalizing flows on categorical tasks, especially for such with non-sequential data.

ACKNOWLEDGEMENTS

We thank SURFsara for the support in using the Lisa Compute Cluster.

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

## A  VISUALIZATIONS AND DETAILS ON ENCODING DISTRIBUTIONS

In the following, we visualize the different encoding distributions we tested in Categorical Normalizing Flows, and outline implementation details for full reproducibility.

### A.1  MIXTURE OF LOGISTICS

The mixture model represents each category by an independent logistic distribution in continuous latent space, as visualized in Figure 5. Specifically, the encoder distribution $q(\boldsymbol{z}|\boldsymbol{x})$, with $\boldsymbol{x}$ being the categorical input and $\boldsymbol{z}$ the continuous latent representation, can be written as:

$$q(\boldsymbol{z}|\boldsymbol{x}) \quad = \quad \prod_{i=1}^{N} g(\boldsymbol{z}_i|\mu(x_i), \sigma(x_i)) \tag{8}$$

$$g(\boldsymbol{v}|\mu, \sigma) \quad = \quad \prod_{j=1}^{d} \frac{\exp(-\epsilon_j)}{(1 + \exp(-\epsilon_j))^2} \quad \text{where} \quad \epsilon_j = \frac{v_j - \mu_j}{\sigma_j} \tag{9}$$

$g$ represent the logistic distribution, and $d$ the dimensionality of the continuous latent space per category. Both parameters $\mu$ and $\sigma$ are learnable parameter, which can be implemented via a simple table lookup. For decoding the discrete categorical data from continuous space, the true posterior is calculated by applying the Bayes rule:

$$p(x_i|\boldsymbol{z}_i) = \frac{\tilde{p}(x_i)q(\boldsymbol{z}_i|x_i)}{\sum_{\hat{x}} \tilde{p}(\hat{x})q(\boldsymbol{z}_i|\hat{x})} \tag{10}$$

where the prior over categories, $\tilde{p}(x_i)$, is calculated based on the category frequencies in the training dataset. Although the posterior models a distribution over categories, the distribution is strongly peaked for most continuous points in the latent space as the probability steeply decreases the further a point is away from a specific mode. Furthermore, the distribution is trained to minimize the posterior entropy which pushes the posterior to be deterministic for commonly sampled continuous points. Hence, the posterior partitions the latent space into fragments in which all continuous points are assigned to one discrete category. The borders between the fragments, where the posterior is not close to deterministic, are small and very rarely sampled by the encoder distribution. We visualized the partitioning for an example of three categories in Figure 5.

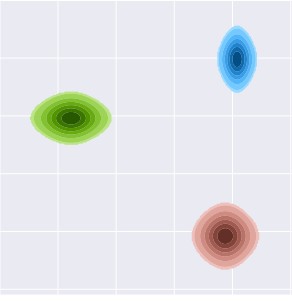
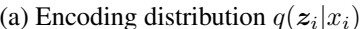
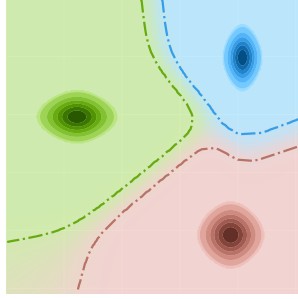

(a) Encoding distribution $q(\boldsymbol{z}_i|x_i)$              (b) Decoder partitioning $p(x_i|\boldsymbol{z_i})$

Figure 5: Visualization of the mixture model encoding and decoding for 3 categories. Best viewed in color. (a) Each category is represented by a logistic distribution with independent mean and scale which are learned during training. (b) The posterior partitions the latent space which we visualize by the background color. The borders show from when on we have an almost unique decoding of the corresponding mixture ($> 0.95$ decoding probability). Note that these borders do not directly correspond to the euclidean distance as we use logistic distributions instead of Gaussians.

Notably, the posterior can also be learned by a second, small linear network. While this possibly introduces a difference between encoder and decoder, we experienced it to vanish quickly over training iterations and did not observe any significant difference compared to using the Bayes posterior besides a slower training in the very early stages of training. Additionally, we were able

to achieve very low reconstruction errors in two dimensions for most discrete distributions of $\leq 50$ categories. Nevertheless, higher dimensionality of the latent space is not only crucial for a large number of categories as for word-level vocabularies but can also be beneficial for more complex problems. Still, using even higher dimensionality rarely caused any problems or showed significantly decreasing performance. Presumably, the flow learns to ignore latent dimensions if those are not needed for modeling the discrete distribution. To summarize, the dimensionality of the latent space should be considered as important, but robust hyperparameter which can be tuned in an early stage of hyperparameter search.

In the very first training iterations, it can happen that the mixtures of multiple categories are at the exact same spot and have troubles to clearly separate. This can be easily resolved by either weighing the reconstruction loss slightly higher for the first $\sim$500 iterations or initializing the mean of the mixtures with a higher variance. Once the mixtures are separated, the model has no incentive to group them together again as it has started to learn the underlying discrete distribution which results in a considerably higher likelihood than a plain uniform distribution.

### A.1.1 KL DIVERGENCE

The objective for learning categorical normalizing flows with a mixture encoding, shown in Equation 3, constitutes a lower bound. The variational gap between the objective and the evidence $\log P_{\text{model}}$ is given by the KL divergence between the approximate posterior $q(\boldsymbol{z}|\boldsymbol{x})$, and the true posterior $p(\boldsymbol{z}|\boldsymbol{x})$: $D_{KL}\left(q(\boldsymbol{z}|\boldsymbol{x})||p(\boldsymbol{z}|\boldsymbol{x})\right)$. The advantage of using the mixture encoding is that by replacing the decoder by a Bayes formulation of the encoder, as in Equation 10, we introduce a dependency between $q(\boldsymbol{z}|\boldsymbol{x})$ and $p(\boldsymbol{z}|\boldsymbol{x})$. Specifically, we can rewrite the true posterior as:

$$p(\boldsymbol{z}|\boldsymbol{x}) \quad = \quad \frac{p(\boldsymbol{x}|\boldsymbol{z})p(\boldsymbol{z})}{p(\boldsymbol{x})} \tag{11}$$

$$= \quad \frac{\prod_i p(x_i|\boldsymbol{z}_i)p(\boldsymbol{z})}{p(\boldsymbol{x})} \tag{12}$$

$$= \quad \left(\prod_i \frac{q(\boldsymbol{z}_i|x_i)p(x_i)}{\sum_{\hat{x}} q(\boldsymbol{z}_i|\hat{x})p(\hat{x})}\right) \frac{p(\boldsymbol{z})}{p(\boldsymbol{x})} \tag{13}$$

$$= \quad \prod_i q(\boldsymbol{z}_i|x_i) \cdot \frac{\prod_i p(x_i)}{p(\boldsymbol{x})} \cdot \frac{p(\boldsymbol{z})}{\prod_i q(\boldsymbol{z}_i)} \tag{14}$$

$$= \quad q(\boldsymbol{z}|\boldsymbol{x}) \cdot \frac{p(\boldsymbol{z})}{p(\boldsymbol{x})} \cdot \prod_i \frac{p(x_i)}{q(\boldsymbol{z}_i)} \tag{15}$$

Intuitively, this makes it easier for the model to tighten the gap because a change in $q(\boldsymbol{z}|\boldsymbol{x})$ entails a change in $p(\boldsymbol{z}|\boldsymbol{x})$. In experiments, we observe the difference by a considerably faster optimization in the first steps during iteration. However, once the decoder is close to deterministic, both approaches with the Bayes formulation and the separate network will reach a similar variational gap as $p(\boldsymbol{x}|\boldsymbol{z})$ will drop out of the Equation 11 with being 1.

Note that this variational gap is very similar to that from variational dequantization for integers, which is being used in most SOTA image modeling architectures. The difference to dequantization is that the decoder $p(x_i|\boldsymbol{z}_i)$ has been manually fixed, but yet represents the Bayes formulation of the encoder $q(\boldsymbol{z}|\boldsymbol{x})$ (the latent variable $\boldsymbol{z}$ represents $\boldsymbol{x} + \boldsymbol{u}$ here, i.e. the original integers with a random variable $\boldsymbol{u}$ between 0 and 1).

### A.1.2 TRAINING AND TESTING

Below, we lay out the specifics for training and testing a categorical normalizing flow with the logistic mixture encoding. Training and testing are almost identical to normalizing flows trained on image modeling, except for the loss calculation and encoding. Algorithm 1 shows the training procedure. First of all, we determine the prior $p(x_i)$ over categories, which can be done by counting the number of occurrences of each category and divide by the sum of all. The difference between the prior probabilities in the training and testing is usually neglectable, while the training set commonly is

larger and hence provides a better overall data statistic. After this, the training can start by iterating over batches in the training set $\mathcal{D}$. For encoding, we can make use of the reparameterization trick and simply shift and scale the samples of a standard logistic distribution. The loss is the lower bound of Equation 3.

---

**Algorithm 1** Training procedure for the logistic mixture encoding in CNFs

1: Calculate prior probabilities $p(x_i)$ on the training dataset;
2: **for** $x \in \mathcal{D}$ **do**
3:     Sample a random logistic variable $z'$: $z' \sim \text{LogisticDist}(0, 1)$;
4:     Reparameterization trick for encoding: $z_i = z_i' \cdot \sigma(x_i) + \mu(x_i)$
5:     Negative log-likelihood calculation $\mathcal{L} = -\log\left(p_{\text{model}}(z) \prod_{i=1}^{S} \frac{\tilde{p}(x_i)}{\sum_{\hat{x}} \tilde{p}(\hat{x}) q(z_i|\hat{x})}\right)$ (Eq. 3)
6:     Minimize loss $\mathcal{L}$ by updating parameters in $p_{\text{model}}(z)$ and $q(z_i|x_i)$;
7: **end for**

---

During testing, we make use of importance sampling to tighten the gap, given that $\log \mathbb{E}_x\left[p(x)\right] \geq \mathbb{E}_x\left[\log \frac{1}{N} \sum_{n=1}^{N} p(x_n)\right] \geq \mathbb{E}_x\left[\log p(x)\right]$ (Burda et al., 2016). This is again a standard technique for evaluating normalizing flows on images, and can improve the bits per dimension score slightly. In our experiments however, we did not experience a significant difference between $N = 1$ and $N = 1000$. The bits per dimension score is calculated by using the log with base 2 on the likelihood, and divide it by the number of dimensions/elements in categorical space (denoted by $S$).

---

**Algorithm 2** Test procedure for the logistic mixture encoding in CNFs. We use $N = 1000$ in our experiments, although the difference between $N = 1$ and $N = 1000$ was marginal for most cases.

1: **for** $x \in \mathcal{D}$ **do**
2:     **for** $n = 1, ..., N$ **do**
3:         Encode $x$ as continuous variable $z$: $z \sim q(z|x)$;
4:         Determine likelihood $\mathcal{L}_n = p_{\text{model}}(z) \prod_i \frac{\tilde{p}(x_i)}{\sum_{\hat{x}} \tilde{p}(\hat{x}) q(z_i|\hat{x})}$;
5:     **end for**
6:     Determine bits per dimension score: $-\log_2\left[\frac{1}{N} \sum_{n=1}^{N} \mathcal{L}_n\right]/S$;
7: **end for**

---

### A.1.3 EXAMPLE ENCODING ON MOLECULE GENERATION

An example of a trained model encoding is shown in Figure 6. Here, we visualize the encoding of the edge attributes in GraphCNF trained the molecule generation. In this setup, we have 3 categories, representing the single, double and triple bond. While the single bond category is clearly the dominant one due to the higher prior probability, we did not observe any specific trends across trained models of the position or scale of the distributions.

Visualizations on graph coloring show similar behavior as Figure 5 because all three categories have the same prior probability. Other encoding distributions such as the node types (atoms) cannot be so easily visualized because of their higher dimensionality than 2. We tried applying dimensionality reduction techniques for those but experienced that those do not capture the distribution shape well.

### A.2 LINEAR FLOWS

The flexibility of the mixture model can be increased by applying normalizing flows on each mixture that dependent on the discrete category. We refer to this approach as *linear flows* as the flows are applied for each categorical input variable independently. We visualize possible encoding distributions

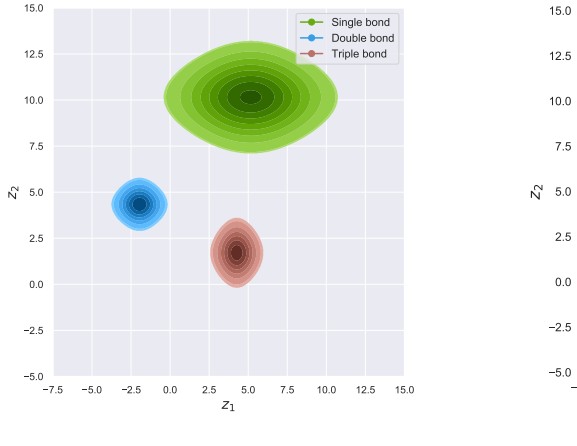 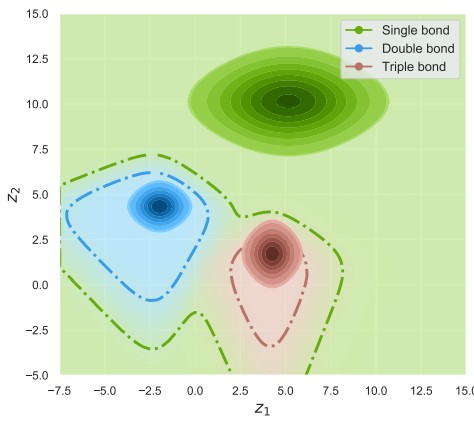

(a) Encoding distribution $q(\boldsymbol{z}_i|x_i)$        (b) Decoder partitioning $p(x_i|\boldsymbol{z_i})$

Figure 6: Visualization of the mixture model encoding of the edge attributes for a trained model on molecule generation. The majority of the space is assigned to category 1, the single bond, as it is by far the most common edge type. Across multiple models however, we did not see a consistent trend of position/standard deviation of each category's encoding.

with linear flows in Figure 7. Formally, we can write the distribution as:

$$q(\boldsymbol{z}|\boldsymbol{x}) \;\;=\;\; \prod_{i=1}^{N} q(\boldsymbol{z}_i|x_i) \tag{16}$$

$$q\left(\boldsymbol{z}^{(K)}\big|x_i\right) \;\;=\;\; g\left(\boldsymbol{z}^{(0)}\right) \cdot \prod_{k=1}^{K} \left|\det \frac{\partial f_k(\boldsymbol{z}^{(k-1)}; x_i)}{\partial \boldsymbol{z}^{(k-1)}}\right| \;\; \text{where} \;\; \boldsymbol{z}_i = \boldsymbol{z}^{(K)} \tag{17}$$

where $f_1, ..., f_K$ are invertible, smooth mappings. In particular, we use here again a sequence of coupling layers with activation normalization and invertible 1x1 convolutions (Kingma and Dhariwal, 2018). Both the activation normalization and coupling use the category $x_i$ as additional external input to determine their transformation parameters by a neural network. The class-conditional transformations could also be implemented by storing $K$ parameter sets for the coupling layer neural networks, which is however inefficient for a larger number of categories. Furthermore, in coupling layers, we apply a channel mask that splits $\boldsymbol{z}_i$ over latent dimensionality $d$ into two equally sized parts, of which one is transformed using the other as input.

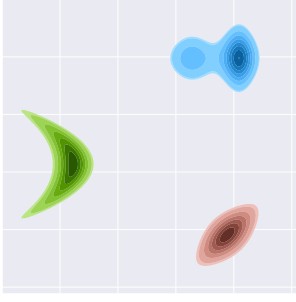

(a) Encoding distribution $q(\boldsymbol{z}_i|x_i)$        (b) Decoder partitioning $p(x_i|\boldsymbol{z_i})$

Figure 7: Visualization of the linear flow encoding and decoding for 3 categories. Best viewed in color. (a) The distribution per category is not restricted to a simple logistic and can be multi-modal, rotated or transformed even more. (b) The posterior partitions the latent space which we visualize by the background color. The borders show from when on we have an almost unique decoding of the corresponding category distribution ($> 0.95$ decoding probability).

Similarly to the mixture model, we can calculate the true posterior $p(x_i|z_i)$ using the Bayes rule. Thereby, we sample from the flow for $x_i$ and need to inverse the flows for all other categories. Note that as the inverse of the flow also needs to be differentiable in this situation, we apply affine coupling layers instead of logistic mixture layers. However, this gets computationally expensive for more than 20 categories, and thus we used a single-layer linear network as posterior in these situations. The partitions of the latent space that can be learned by the encoding distribution are much more flexible, as illustrated in Figure 7.

We experimented with increasing sizes of linear flows but noticed that the encoding distribution usually fell back to rotated logistic distributions. The fact that the added complexity and flexibility by the flows are not being used further supports our observation that mixture models are indeed sufficient for representing categorical data well in normalizing flows.

### A.3    VARIATIONAL ENCODING

The third encoding distribution we experimented with is inspired by variational dequantization (Ho et al., 2019) and models $q(z|x)$ by one flow across all categorical variables. Still, the posterior, $p(x_i|z_i)$, is applied per categorical variable independently to maintain unique decoding and partitioning of the latent space. The normalizing flow again consists of a sequence of logistic mixture coupling layers with activation normalization and invertible 1x1 convolutions. The inner feature network of the coupling layers depends on the task the normalizing flow is applied on. Hence, for sets, we used a transformer architecture, while for the graph experiments, we used a GNN. On the language modeling task, we used a Bi-LSTM model to generate the transformation parameters. All those networks use the discrete, categorical data $x$ as additional input.

As the true posterior cannot be found for this distribution, we apply a two-layer linear network to determine $p(x_i|z_i)$. While the reconstruction error was again very low, we again experienced that the model mainly relied on a logistic mixture model, even if we initialize it differently beforehand. Variational dequantization is presumably important for images as every pixel value has its own independent Gaussian noise signal. This noise can be nicely modeled by flexible dequantization distributions which need to be complex enough to capture the true mean and variance of this Gaussian noise. In categorical distributions, however, we do not have such noise signals and therefore seem not to benefit from variational encodings.

### A.4    LATENT NORMALIZING FLOW

#### A.4.1    ENCODING VISUALIZATION

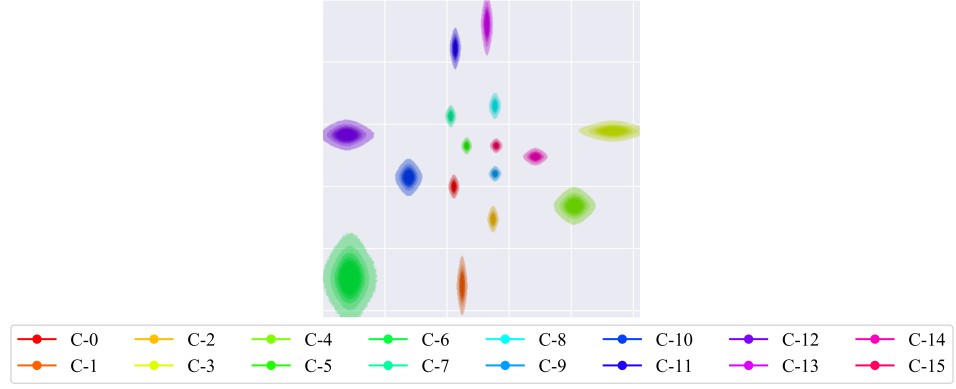

Figure 8: Visualization of the encoding distribution $q(z|x)$ of latent NF on set summation.

In this section, we show visualizations of the learned encoding distribution of Latent NF (Ziegler and Rush, 2019) on the task of set summation. As shown in Figure 8, the encoder learned a clear mixture distribution although not being restricted too. This shows that the possible complexity in the decoder is not being used. The figure was generated by sampling $5,000$ examples and merging them into one plot. Each latent variable $z_i$ is two dimensional, hence our two-dimensional plot. The

colors represent different categories. For readability, each color is normalized independently (i.e. highest value 1, lowest 0) as otherwise, the colors of the stretched mixtures would be too low. For visualizations of the latent space in language modeling, we refer the reader to Ziegler and Rush (2019).

### A.4.2 Loss comparison

In the following, we compare LatentNF and CNF on their training behavior and loss fluctuation. Figure 9 visualizes the loss curve for both models on the task of language modeling, trained on the text8 dataset (Mahoney, 2011). Note that both models use the same hyperparameters and model except that the reconstruction loss is weighted 10 times higher in the first 5k iterations, and decays exponentially over consecutive iterations. This is why the initial loss of LatentNF is higher, but the reconstruction loss becomes close to zero after 3k iterations in the example of Figure 9. Interestingly, we experience high fluctuations of the loss with LatentNF during the first iterations. Although Figure 9 shows only a single run, similar fluctuations have occurred in all trained LatentNF models but at different training iterations. Hence, we decided to plot a single loss curve instead of the average of multiple. The fluctuations can most likely be explained by the need for a strong loss scheduling. At the beginning of the training, the decoder loss is weighted significantly higher than the prior component. Thus, the backpropagated gradients mostly focus on the decoder, which can peak for rare categories/occurrences in the dataset. These peaks cause the encoding distribution to change abruptly, and the prior has to adapt to these changes within the next iterations. However, this can again lead to a high loss when the transformations in the prior do not fit to the new encoding, and thus map them to points of low likelihood. Thus, we see a loss peak across a couple of iterations until the model balances itself out again.

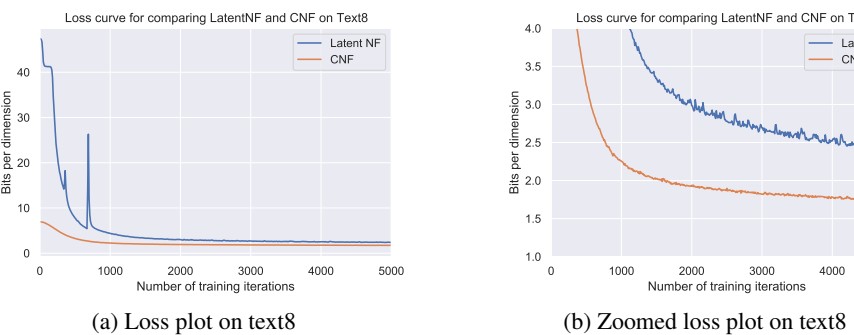

(a) Loss plot on text8          (b) Zoomed loss plot on text8

Figure 9: Plotting the loss over batch iterations on the language modeling dataset text8 for LatentNF and CNF. The batch size is 128, and we average the loss over the last 10 iterations. Sub-figure (b) shows a zoomed version of sub-figure (a) to show the larger fluctuation even on smaller scale, while CNF provides a smooth optimization.

Similarly, when training on Wikitext103 (Merity et al., 2017), we experience even more frequent peaks in the loss, as Wikitext103 with 10k categories contains even more rare occurrences (see Figure 10). Reducing the learning rate did not show to improve the stability while considerably increasing the training time. When reducing the weight of the decoder, the model was not able to optimize as well as before and usually reached bits per dimension scores of 10-20.

## B Implementation details of GraphCNF

In this section, we describe further implementation details of GraphCNF. We detail the implementation of the Edge-GNN model used in the coupling layers of GraphCNF, and discuss how we encode graphs of different sizes.

### B.1 Edge Graph Neural Network

GraphCNF implements a three-step generation approach, for which the second and third step also models latent variables for edges. Hence, in the coupling layers, we need a graph neural network

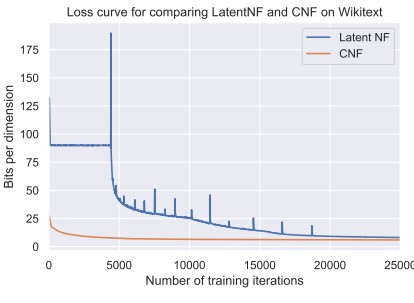

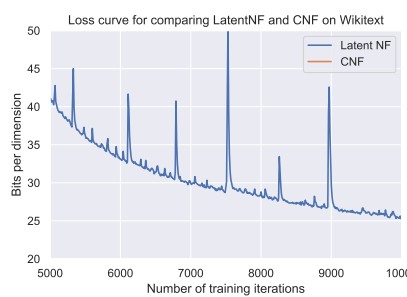

(a) Loss plot on Wikitext103

(b) Zoomed loss plot on Wikitext103

Figure 10: Plotting the loss over batch iterations on the language modeling dataset Wikitext103 for LatentNF and CNF. The batch size is 128, and we average the loss over the last 10 iterations. Sub-figure (b) shows a zoomed version of sub-figure (a) to show the details of the peaks in LatentNF's loss curve.

which supports both node and edge features. We implement this by alternating between updates of the edge and the node features. Specifically, given node features $\boldsymbol{v}^t$ and edge features $\boldsymbol{e}^t$ at layer $t$, we update those as follows:

$$
\begin{aligned}
\boldsymbol{v}^{t+1} &= f_{node}(\boldsymbol{v}^t; \boldsymbol{e}^t) & (18) \\
\boldsymbol{e}^{t+1} &= f_{edge}(\boldsymbol{e}^t; \boldsymbol{v}^{t+1}) & (19)
\end{aligned}
$$

The update functions, $f_{node}$ and $f_{edge}$, are both common GNN layers with slight adjustments to allow a communication between nodes and edges. Before detailing the update layers, it should be noted that we use Highway GNNs (Rahimi et al., 2018) which apply a gating mechanism. Specifically, the updates for the nodes are determined by:

$$
\boldsymbol{v}^{t+1} = \boldsymbol{v}^t \cdot T\left(\tilde{\boldsymbol{v}}^{t+1}\right) + H\left(\tilde{\boldsymbol{v}}^{t+1}\right) \cdot \left(1 - T\left(\tilde{\boldsymbol{v}}^{t+1}\right)\right) \tag{20}
$$

where $\tilde{\boldsymbol{v}}^{t+1}$ is the output of the GNN layer. $H$ and $T$ represent single linear layer networks where $T$ has a consecutive sigmoid activation to limit the outputs between 0 and 1. The edge updates are applied in the similar manner. We experienced that such a gated update functions helps the gradient flow through the layers back to the input. This is important for normalizing flows as coupling layers or transformations in general strongly depend on previous transformations. Hence, we apply the same gating mechanism in the first step of GraphCNF, $f_1$.

Next, we detail the GNN layers to obtain $\tilde{e}^{t+1}$ and $\tilde{v}^{t+1}$. The edge update layer $f_{edge}$ resembles a graph convolutional layer (Zhou et al., 2018), and can be specified as follows:

$$
\tilde{e}_{ij}^{t+1} = g\left(W_e^t e_{ij}^t + W_v^t v_i^t + W_v^t v_j^t\right) \tag{21}
$$

where $e_{ij}$ represents the features of the edge between node $i$ and $j$. $g$ stands for a GELU (Hendrycks and Gimpel, 2016) non-linear activation. Using more complex transformations did not show to significantly improve the performance of GraphCNF.

To update the node representations, we took inspiration of the transformer architecture (Vaswani et al., 2017) and use a modified multi-head attention layer. In particular, a linear transformation maps each node to a key, query and value vector:

$$
K_{v_i}, Q_{v_i}, V_{v_i} = W_K v_i^t, W_Q v_i^t, W_V v_i^t \tag{22}
$$

The attention value is usually computed based on the dot product between two nodes. However, as we explicitly have features for the edge between the two nodes, we use those to control the attention mechanism. Hence, we have an additional weight matrix $u$ to map the edge features to an attention bias:

$$
\hat{a}_{ij} = Q_{v_i} K_{v_i}^T / \sqrt{d} + e_{ij}^{t+1} u^T \tag{23}
$$

where $d$ represents the hidden dimensionality of the features. Finally, we also add a edge-based value vector to allow a full communication from edges to nodes. Overall, the updates node features are

calculated by:

$$a_{ij} \quad = \quad \frac{\exp\left(\hat{a}_{ij}\right)}{\sum_m \exp\left(\hat{a}_{im}\right)}, \tag{24}$$

$$\tilde{v}_i^{t+1} \quad = \quad \sum_j a_{ij} \cdot \left[V_{v_j} + W_e e_{ij}^{t+1}\right] \tag{25}$$

Alternatively to transformers, we also experimented with Graph Attention Networks (Veličković et al., 2018). However, those showed slightly worse results which is why we used the transformer-based layer.

In step 2, the (binary) adjacency matrix is given such that each node has a limited number of neighbors. A full transformer-based architecture as above is then not necessary anymore as every atom has usually between 1 and 3 neighbors. Especially the node-to-node dot product is expensive to perform. Hence, we experimented with a node update layer where the attention is purely based on the edge features in step 2. We found both to work equally well while the second is computationally more efficient.

## B.2 ENCODING GRAPH SIZE

The number of nodes $N$ varies across graphs in the dataset, and hence a generative model needs to be flexible regarding $N$. To encode the number of nodes, we use a similar approach as Ziegler and Rush (2019) for sequences and add a simple prior over $N$. The prior is parameterized based on the graph size frequency in the training set. Alternatively, to integrate the number of nodes in the latent space, we could add *virtual* nodes to the model, similar to virtual edges. Every graph in the training dataset would be filled up to the maximum number of nodes (38 for Zinc250k (Irwin et al., 2012)) by adding such virtual nodes. Meanwhile, during sampling, we remove virtual nodes if the model generates such. GraphNVP (Madhawa et al., 2019) uses such an encoding as their coupling layers did not support flexible graph sizes. However, in experiments, we obtained similar performance with both size encodings while the external prior is computationally more efficient and therefore used in this paper.

## C ADDITIONAL RESULTS ON MOLECULE GENERATION

In this section, we present additional results on the molecule generation task. Table 4 shows the results of our model on the Zinc250k (Irwin et al., 2012) dataset including the likelihood on the test set in bits per node. We calculate this metric by summing the log-likelihood of all latent variables, both nodes, and edges, and divide by the number of nodes. Although the number of edges scales with $\mathcal{O}(N^2)$, a higher proportion of those are virtual and did not have a significant contribution to the likelihood. Thus, bits per node constitutes a good metric for comparing the likelihood of molecules of varying size. Additionally, we also report the standard deviation for all metrics over 4 independent runs. For this, we initialized the random number generator with the seeds 42, 43, 44, and 45 before creating the model. The specific validity values we obtained are 80.74%, 81.16%, 85.3% and 86.44% (in no particular order). It should be noted that the standard deviation among those models is considerably high. This is because the models in molecule generation are trained on maximizing the likelihood of the training dataset and not explicitly on generating valid molecules. We experienced that among over seeds, models that perform better in terms of likelihood do not necessarily perform better in terms of validity.

Table 4: Performance on molecule generation trained on Zinc250k (Irwin et al., 2012) with standard deviation is calculated over 4 independent runs. See Table 1 for baselines.

| Method | Validity | Uniqueness | Novelty | Reconstruction | Bits per node |
|---|---|---|---|---|---|
| GraphCNF | 83.41% | 99.99% | 100% | 100% | 5.17bpd |
|  | (±2.88) | (±0.01) | (±0.00) | (±0.00) | (±0.05) |
| + Sub-graphs | 96.35% | 99.98% | 99.98% | 100% |  |
|  | (±2.21) | (±0.01) | (±0.02) | (±0.00) |  |

We also evaluated GraphCNF on the Moses (Polykovskiy et al., 2018) molecule dataset. Moses contains 1.9 million molecules with up to 30 heavy atoms of 7 different types. Again, we follow the preprocessing of Shi et al. (2020) and represent molecules in kekulized form in which hydrogen is removed. The results can be found in Table 5 and show that we achieve very similar scores to the experiments on Zinc250k. Compared to the normalizing flow baseline GraphAF, GraphCNF generates considerably more valid atoms while being parallel in generation in contrast to GraphAF being autoregressive. JT-VAE uses manually encoded rules for generating valid molecules only such that the validity rate is 100%. Overall, the experiment on Moses validates that GraphCNF is not specialized on a single dataset but can improve on current flow-based graph models across datasets.

Table 5: Performance on molecule generation Moses (Polykovskiy et al., 2018), calculated on 10k samples and averaged over 4 runs. Score for GraphAF taken from Shi et al. (2020), and JT-VAE from Polykovskiy et al. (2018).

| Method | Validity | Uniqueness | Novelty | Bits per node |
|---|---|---|---|---|
| JT-VAE (Jin et al., 2018) | 100% | 99.92% | 91.53% | - |
| GraphAF (Shi et al., 2020) | 71% | 99.99% | 100% | - |
| GraphCNF | 82.56% | 100.0% | 100% | 4.94bpd |
| | ($\pm 2.34$) | ($\pm 0.00$) | ($\pm 0.00$) | ($\pm 0.04$) |
| + Sub-graphs | 95.66% | 99.98% | 100% | |
| | ($\pm 2.58$) | ($\pm 0.01$) | ($\pm 0.00$) | |

Finally, we show 12 randomly sampled molecules from our model in Figure 11. In general, GraphCNF is able to generate a very diverse set of molecules with a variety of atom types. This qualitative analysis endorses the previous quantitative results of obtaining close to 100% uniqueness on 10k samples.

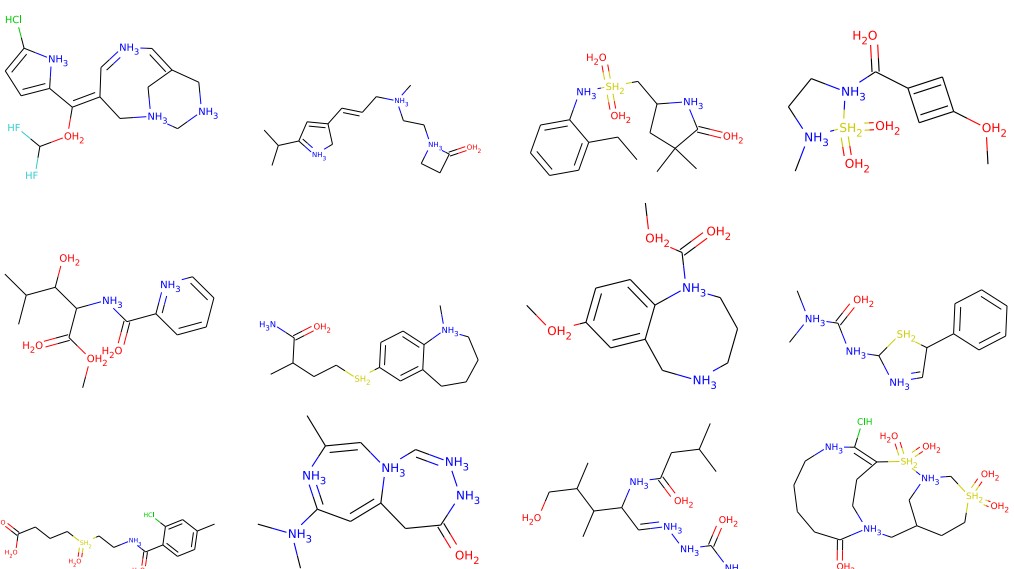

Figure 11: Visualization of molecules generated by GraphCNF which has been trained on the Zinc250k (Irwin et al., 2012) dataset. Nodes with black connections and no description represent carbon atoms. All of the presented molecules are valid. Best viewed in color and electronically for large molecules.

Besides generating multiple sub-graphs, the most common failure case we have found are single, invalid edges in a large molecule, as shown in four examples in Figure 12.

Figure 12: Failure cases in molecule generation besides sub-graph generation. The invalid edges have been indicates with a green arrow. Changing the edges to single bonds in the examples above would constitute a valid molecule.

## D  EXPERIMENTAL SETTINGS

In this section, we detail the hyperparameter settings and datasets for all experiments. All experiments have been implemented using the deep learning framework PyTorch (Paszke et al., 2019). The experiments for graph coloring and molecule generation have been executed on a single NVIDIA TitanRTX GPU. The average training time was between 1 and 2 days. The set and language experiments have been executed on a single NVIDIA GTX1080Ti in 4 to 16 hours. All experiments have been repeated with at least 3 different random seeds.

### D.1  SET MODELING

**Dataset details**  We use two toy datasets, set shuffling and set summation, to simulate a discrete distribution over sets in our experiments. Note that we do not have a classical split of train/val/test dataset, but instead train and test the models on samples from the same discrete distribution. This is because we want to verify whether a categorical normalizing flow and other baselines can model an arbitrary discrete distribution. The special property of sets is that permuting the elements of a set still represent the same set. However, a generative model still has to learn all possible permutations. While an autoregressive model considers those permutations as different data points, a permutation-invariant model as Categorical Normalizing Flow contains an inductive bias to assign the same likelihood to any permutation.

In set shuffling, we only have one set to model which is the following (with categories $C_1$ to $C_{16}$):

$$\{C_1, C_2, C_3, C_4, C_5, C_6, C_7, C_8, C_9, C_{10}, C_{11}, C_{12}, C_{13}, C_{14}, C_{15}, C_{16}\}$$

This set has 16! possible permutations and therefore challenging to model. The optimal likelihood in bits per element is calculated by $\log_2 (16!) / 16 \approx 2.77$.

The dataset set summing contains of 2200 valid sets for $N = 16$ and $L = 42$. An example for a valid set is:

$$\{1, 1, 1, 1, 2, 2, 2, 2, 2, 2, 3, 3, 3, 3, 6, 8\}$$

For readability, the set is sorted by ascending values, although any permutation of the elements represent the exact same set. Taking into account all possible permutations of the sets in the dataset, we obtain a optimal likelihood of $\log_2 (6.3 \cdot 10^{10}) / 16 \approx 2.24$. The values for the sequence length $N$ and sum $L$ was chosen such that the task is challenging enough to show the differences between Categorical Normalizing Flows and its baselines, but also not too challenging to prevent unnecessarily long training times and model complexities.

**Hyperparameter details**  Table 6 shows an overview of the hyperparameters per model applied on set modeling. We use the notation "{val1, val2, ...}" to show the different values we have tried during hyperparameter search. Thereby, the underlined value denotes the hyperparameter value with the best performance and finally was being used to generate the results in Table 3.

The number of encoding coupling layers in Categorical Normalizing Flows are sorted by the used encoding distribution. The mixture model uses no additional coupling layers, while for the linear flows, we apply 4 affine coupling layers using an external input for the discrete category. For the variational encoding distribution $q(z|x)$, we use 4 mixture coupling layers across the all latent variables $z$ with external input for $x$. A larger dimensionality of the latent space per element showed to be beneficial for all encoding distributions. Note that due to a dimensionality larger than 1 per

element, we can apply the channel mask instead of a chess mask and maintain permutation invariance compared to the baselines.

In variational dequantization and Discrete NF, we sort the categories randomly for set shuffling (the distribution is invariant to the category order/assignment) and in ascending order for set summation. In Discrete NF, we followed the published code from Tran et al. (2019) for their coupling layers and implemented it in PyTorch (Paszke et al., 2019). We use a discrete prior over the set elements which is jointly optimized with the flow. However, we experienced significant optimization issues due to the straight-through gradient estimator in the Gumbel Softmax.

Across this paper, we experiment with the two optimizers Adam (Kingma and Ba, 2015) and RAdam (Liu et al., 2020), and experienced RAdam to work slightly better. The learning rate decay is applied every update and leads to exponential decay. However, we did not observe the choice of this hyperparameter to be crucial.

Table 6: Hyperparameter overview for the set modeling experiments presented in Table 3

| Hyperparameters | Categorical NF | Var. dequant. | Discrete NF |
|---|---|---|---|
| Latent dimension | $\{2, \underline{4}, 6\}$ | 1 | 16 |
| #Encoding couplings | - / 4 / 4 | 4 | - |
| #Coupling layers | 8 | 8 | $\{4, \underline{8}\}$ |
| Coupling network | Transformer | Transformer | Transformer |
| - Number of layers | 2 | 2 | 2 |
| - Hidden size | 256 | 256 | 256 |
| Mask | Channel mask | Chess mask | Chess mask |
| #mixtures | 8 | 8 | - |
| Batch size | 1024 | 1024 | 1024 |
| Training iterations | 100k | 100k | 100k |
| Optimizer | $\{$Adam, $\underline{RAdam}\}$ | RAdam | $\{$SGD, $\underline{Adam}$, RAdam$\}$ |
| Learning rate | 7.5e-4 | 7.5e-4 | $\{$1e-3, $\underline{1e-4}$, 1e-5$\}$ |
| Learning rate decay | 0.999975 | 0.999975 | 0.999975 |
| Temperature (GS) | - | - | $\{\underline{0.1}, 0.2, 0.5\}$ |

## D.2 GRAPH COLORING

**Dataset details** In our experiments, we focus on the 3-color problem meaning that a graph has to be colored using $K = 3$ colors. We generate the datasets by randomly sampling a graph and using an SAT solver[2] for finding one valid coloring assignment. In case no solution can be found, we discard the graph and sample a new graph. We further ensure that every graph cannot be colored by less than 3 colors to exclude too simple graphs. For creating the graphs, we take inspiration from Lemos et al. (2019) and first uniformly sample the number of nodes between $10 \leq |V| \leq 20$ for the small dataset, and $25 \leq |V| \leq 50$ for the large dataset. Next, we sample a value $p$ between 0.1 and 0.3 which represents the probability of having an edge between a random pair of nodes. Thus, $p$ controls how dense a graph is, and we aim to have both dense and sparse graphs in our dataset. Finally, for each pair of nodes, we sample from a Bernoulli distribution with probability $p$ of adding an edge between the two nodes or not. Finally, we check whether each node has at least one connection and that all nodes can be reached from any other node. This ensures that we have one connected graph and not multiple sub-graphs. Overall, we create a train/val/test size of 192k/24k/24k for the small dataset, and 450k/20k/30k for the large graphs. We visualize examples of the datasets in Figure 13.

During training, we randomly permute the colors of a graph (e.g. red becomes blue, blue becomes green, green becomes red) as any permutation is a valid color assignment. When we sample a color assignment from our models, we explicitly use a temperature value of 1.0. For the autoregressive

---

[2]We have used the following solver from the OR-Tools library in python: https://developers.google.com/optimization/cp/cp_solver

model and the VAE, this means that we sample from the softmax output. A common alternative is to take the argmax, which corresponds to a temperature value of 0.0. However, we stick to the original distribution because we want to test whether the models capture the full discrete distribution of valid color assignments and not only the most likely solution. For the normalizing flow, a temperature of 1.0 corresponds to sampling from the prior distribution as it was used during training.

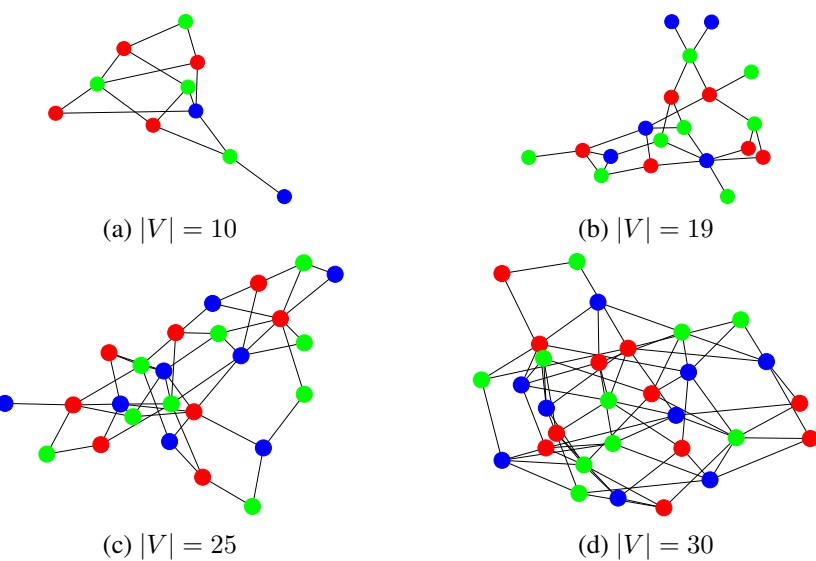

(a) $|V| = 10$

(b) $|V| = 19$

(c) $|V| = 25$

(d) $|V| = 30$

Figure 13: Examples of valid graph color assignments from the dataset (best viewed in color). Due to the graph sizes and dense adjacency matrices, edges can be occluded or cluttered in (c) and (d).

**Hyperparameter details** Table 7 shows an overview of the used hyperparameters. If "/" is used in the table, the first parameter refers to the hyperparameter value used on a small dataset and the second for the larger dataset. The activation function used within the graph neural networks is GELU (Hendrycks and Gimpel, 2016). Interestingly we experience that a larger latent space dimensionality is crucial for larger graphs despite having the same number of categories as the small dataset. This shows that having an encoding being flexible in the number of dimensions can be further important for datasets where complex relations between categorical variables need to be modeled. Increasing the number of dimensions on the small dataset did not show any significant differences in performance. The number of mixtures in the mixture coupling layers is in general beneficial to be large. However, this can also increase the sampling time. In the case of sampling time being crucial, the number of mixtures can be decreased for the tradeoff of slightly worse performance.

The input to the autoregressive model is the graph with the color assignment at time step $T$ where each category including unassigned nodes is represented by an embedding vector. We experiment with an increasing number of hidden layers. While more layers are especially important for sub-optimal node ordering, the performance does not significantly improve for more than 5 layers. As the sampling time also increases linearly with the number of layers, we use 5 hidden layers for the models.

For the variational autoencoder, we encode each node by a latent vector of size 4. As VAEs have shown to benefit from slowly adding the KL divergence between prior and posterior to the loss, we experiment with a scheduler where the slope is based on a sigmoid and stretched over 10k iterations. We apply a 5 layer graph attention network for both the encoder and decoder. Increasing the number of layers did not show a significant gain while making the loss scheduling more difficult, which is why we stuck with 5 layers.

**Detailed results** Table 8 shows the standard deviation of the results reported in Table 2. Each model was run with 3 different seeds.

Table 7: Hyperparameter overview for graph coloring experiments presented in Table 2

| Hyperparameters | GraphCNF | Variational AE | Autoregressive |
|---|---|---|---|
| Latent dimension | {$\underline{2}$, 4} / {2, 4, $\underline{6}$, 8} | 4 | - |
| #Coupling layers | {6, $\underline{8}$} | - | - |
| (Coupling) network | GAT | GAT | GAT |
| - Number of layers | {3, $\underline{4}$, 5} | 5 | {3, 4, $\underline{5}$, 6, 7} |
| - Hidden size | 384 | 384 | 384 |
| - Number of heads | 4 | 4 | 4 |
| Mask | Channel mask | - | - |
| #mixtures | {4, $\underline{8}$, 16} / {4, 8, $\underline{16}$} | - | - |
| Batch size | 384 / 128 | 384 / 128 | 384 / 128 |
| Training iterations | 200k | 200k | 100k |
| Optimizer | RAdam | RAdam | RAdam |
| Learning rate | 7.5e-4 | 7.5e-4 | 7.5e-4 |
| KL scheduler | - | {1.0, $\underline{0.1 \rightarrow 0.5}$, 0.1→1.0} | - |

Table 8: Results on the graph coloring problem, including standard deviation over 3 seeds. The column *time* is excluded since the execution time per batch is constant over seeds.

| Method | $10 \leq \|V\| \leq 20$ | | $25 \leq \|V\| \leq 50$ | |
|---|---|---|---|---|
| | Validity | Bits per node | Validity | Bits per node |
| VAE | 44.95% ±5.32% | 0.84 ±0.04 | 7.75% ±1.59% | 0.64 ±0.02 |
| RNN+Smallest_first | 76.86% ±0.84% | 0.73 ±0.02 | 32.27% ±1.41% | 0.50 ±0.01 |
| RNN+Random | 88.62% ±0.65% | 0.70 ±0.01 | 49.28% ±1.53% | 0.46 ±0.01 |
| RNN+Largest_first | 93.41% ±0.42% | 0.68 ±0.01 | **71.32%** ±0.77% | **0.43** ±0.01 |
| GraphCNF | **94.56%** ±0.55% | **0.67** ±0.00 | 66.80% ±1.14% | 0.45 ±0.01 |
| − Affine | 93.90% ±0.68% | 0.69 ±0.02 | 65.78% ±1.03% | 0.47 ±0.01 |

### D.3 MOLECULE GENERATION

**Dataset details** The Zinc250k (Irwin et al., 2012) dataset we use contains 239k molecules of which we use 214k molecules for training, 8k for validation, and 17k for testing. We follow the preprocessing of Shi et al. (2020) and represent molecules in kekulized form in which hydrogen is removed. This leaves the molecules with up to 38 heavy atoms, with a mean and median size of about 23. The smallest graph consists of 8 nodes. Thereby, Zinc250k considers molecule with 8 different atom types where the distribution is significantly imbalanced. The most common atom is carbon with 73% of all nodes in the dataset. Besides oxygen (10%) and nitrogen (12%), the rest of the atoms occur in less than 2% of all nodes, with the rarest atom being Bromine (0.002%). Between those atoms, the dataset contains 3 different bonds or edge types, namely single, double and triple covalent bonds describing how many electrons are shared among the atoms. In over 90% of all node pairs there exist no bond. In 7% of the cases, the atoms are connected with a single connection, 2.4% with a double, and 0.02% with a triple connection. A similar imbalance is present in the Moses dataset and is based on the properties of molecules. Nevertheless, we experienced that GraphCNF was able to generate a similar distribution, where adding the third stage (adding virtual edges later) considerably helped to stabilize the edge imbalance.

**Hyperparameter details** We summarize our hyperparameters in Table 9. Generally, a higher latent dimensionality is beneficial for representing nodes/atoms, similar to the graph coloring task. However, we experienced that a lower dimensionality for edges is slightly better, presumably because the flow already has a significant amount of latent variables for edges. Many edges, especially the virtual ones, do not contain much information. Besides, a deeper flow showed to gain better results offering more complex transformations. However, in contrast to the graph coloring model, GraphCNF on

molecule generation requires a considerable amount of memory as we have to model a feature vector per edge. Nevertheless, we did not experience any issues due to the limited batch size of 96, and during testing, we could scale up the batch size easily to more than 128 on an NVIDIA GTX 1080Ti for both datasets.

Table 9: Hyperparameter overview for molecule generation experiments presented in Table 1 and 5

| Hyperparameters | GraphCNF |
|---|---|
| Latent dimension (V/E) | $\{4, \underline{6}, 8\}$ / $\{\underline{2}, 3, 4\}$ |
| #Coupling layers ($f_1/f_2/f_3$) | $4$ / $\{4, \underline{6}\}$ / $\{4, \underline{6}\}$ |
| Coupling network ($f_1/f_{2,3}$) | Relational GCN / Edge-GNN |
| - Number of layers ($f_1/f_2/f_3$) | $\{3/3/3, \underline{3/4/4}, 4/4/4\}$ |
| - Hidden size (V/E) | $\{\underline{256}, 384\}$ / $\{\underline{128}, 192\}$ |
| Mask | Channel mask |
| #mixtures (V/E) | $\{8, \underline{16}\}$ / $\{4, \underline{8}, 16\}$ |
| Batch size (Zinc250k/Moses) | 64 / 96 |
| Training iterations | 150k |
| Optimizer | RAdam (Liu et al., 2020) |
| Learning rate | 2e-4, 5e-4, 7.5e-4, 1e-3 |

## D.4 LANGUAGE MODELING

**Dataset details** The three datasets we use for language modeling are the Penn Treebank (Marcus et al., 1994), text8 and Wikitext103 (Merity et al., 2017). The Penn Treebank with a preprocessing of Mikolov et al. (2012) consists of approximately 5M characters and has a vocabulary size of $K = 51$. We follow the setup of Ziegler and Rush (2019) and split the dataset into sentences of a maximum length of 288. Furthermore, instead of an end-of-sentence token, the length is passed to the model and encoded by an external discrete prior which is created based on the sentence lengths in the training dataset.

Text8 contains about 100M characters and has a vocabulary size of $K = 27$. We again follow the preprocessing of Mikolov et al. (2012) and split the dataset into 90M characters for training, and 5M characters each for validation and testing. We train and test the models on a sequence length of 256.

In contrast to the previous two datasets, we use Wikitext103 as a word-level language dataset. First, we create a vocabulary and limit it to the most frequent 10,000 words in the training corpus. We thereby use pre-trained Glove (Pennington et al., 2014) embeddings to represent the words in the baseline LSTM networks and to determine the logistic mixture parameters in the encoding distribution of Categorical Normalizing Flows. Due to this calculation of the mixture parameters, we use a small linear network as a decoder. The linear network consists of three linear layers of hidden size 512 with GELU (Hendrycks and Gimpel, 2016) activation and output size of 10,000 (the vocabulary size). Similarly to text8, we train and test the models on an input sequence length of 256.

**Hyperparameter details** The hyperparameters for the language modeling experiments are summarized in Table 10. We apply the same hyperparameters for the flow and baseline if applicable. The best latent dimensionality for character-level is 3, although larger dimensionality showed to gain similar performance. For the word-level dataset, it is beneficial to increase the latent dimensionality to 10. However, note that 10 is still significantly smaller than the Glove vector size of 300. As Penn Treebank has a limited training dataset on which LSTM networks easily overfit, we use a dropout (Srivastava et al., 2014) of 0.3 throughout the models and dropout an input token with a chance of 0.1. The other datasets seemed to benefit slightly by a small input dropout to prevent overfitting at later stages of the training.

**Detailed results** Table 11 shows the detailed numbers and standard deviations of the results reported in Figure 4. Each model was run with 3 different seeds based on which the standard deviation has been calculated.

Table 10: Hyperparameter overview for the language modeling experiments presented in Table 11

| Hyperparameters | Penn Treebank | text8 | Wikitext103 |
|---|---|---|---|
| (Max) Sequence length | 288 | 256 | 256 |
| Latent dimension | {2, 3, 4} | {2, 3, 4} | {8, 10, 12} |
| #Coupling layers | 1 | 1 | 1 |
| Coupling network | LSTM | LSTM | LSTM |
| - Number of layers | 1 | 2 | 2 |
| - Hidden size | 1024 | 1024 | 1024 |
| - Dropout | {0.0, 0.3} | 0.0 | 0.0 |
| - Input dropout | {0.0, 0.05, 0.1, 0.2} | {0.0, 0.05, 0.1} | {0.0, 0.05, 0.1} |
| #mixtures | 51 | 27 | 64 |
| Batch size | 128 | 128 | 128 |
| Training iterations | 100k | 150k | 150k |
| Optimizer | RAdam | RAdam | RAdam |
| Learning rate | 7.5e-4 | 7.5e-4 | 7.5e-4 |

Table 11: Results on language modeling. The reconstruction error is shown in brackets.

| Model | Penn Treebank | text8 | Wikitext103 |
|---|---|---|---|
| LSTM baseline | 1.28 ±0.01 | 1.44 ±0.01 | 4.81 ±0.05 |
| Latent NF - 1 layer | 1.30±0.01 (0.01) | 1.61±0.02 (0.03) | 6.39±0.19 (1.78) |
| Categorical NF - 1 layer | 1.27 ±0.01 (0.00) | 1.45 ±0.01 (0.00) | 5.43 ±0.09 (0.32) |

## D.5 REAL-WORLD EXPERIMENTS

To verify that Categorical Normalizing Flows can also be applied to real-world data, we have experimented on a credit-card risk record (Dua and Graff, 2019). The data contains different categorical attributes regarding potential credit risk, and we have used the following 9 attributes: "checking_status", "credit_history", "savings_status", "employment", "housing", "job", "own_telephone", "foreign_worker", and "class". The task is to model the joint density function of those 9 attributes over a dataset of 1000 entries. We have used the first 750 entries for training, 100 for validation, and 150 for testing. The results are shown in Table 12. Both the LatentNF and CNF perform equally well, given the small size of the dataset and number of categories. The results have been averaged over three seeds. However, we experienced a relatively high standard deviation due to the small size of the training and test set.

Table 12: Results on credit card risk dataset (Dua and Graff, 2019). The reconstruction error is shown in brackets where applicable.

| Model | Likelihood in bpd |
|---|---|
| Variational Dequantization | 1.95±0.04 |
| Latent NF | 1.36±0.03 (0.01) |
| Categorical NF | 1.37 ±0.03 (0.00) |

