# OpenReview forum: "Categorical Normalizing Flows via Continuous Transformations"
_ICLR.cc/2021/Conference — ICLR 2021 Poster_

### Official Review · AnonReviewer3 · 2020-10-27
**Inspiring formulation to work with categorical data using flows.**

**Rating:** 7
**Confidence:** 3

**Review:**

### Summary:

This work uses the idea of variational inference to map categorical data to continuous space affording the use of normalizing flows. Authors use several ideas to increase their framework's applicability--factorized distribution assumption, use of multi-scale architecture for step-generation, and permutation invariant components—achieving favorable results on several problems. The approach seems to be especially useful when data is non-sequential.

### Strength:

I like the idea; it is simple and seems to be very useful in specific problems, especially those without a natural order on categorical variables. The authors also make an excellent effort to gather empirical evidence in favor of their method. The literature review is comprehensive, and the method is well-placed among the cited papers. Several sections are well written and have a nice natural flow. I find the application to graphs an excellent use of the multi-scale architectures in coupling-based flows and compliment the authors for the superb visualization in Figure 1.

### Concerns:

I found the most critical section in the paper least clearly explained--section 2.2. I appreciate the overall idea of this work and believe I understand the work well, yet the use of ill-defined terms and lack of clarity hamper my confidence.

In my current understanding of the paper, $p(z)$ is modeled using a large flow. Authors use the motivation that $KL[q||p]$ is negligible and approximate $p(x_i|z_i)$ using $q(z_i|x_i)$. With this, eq. 2 can be used to train; however, I am unsure if that is indeed the case; I will appreciate it if the rebuttal absolutely clarifies this.  To make it easier for the authors, I effort to pinpoint the specific instances where they lose me in section 2.2.

In the first paragraph, I find the last line: "Specifically, as normalizing .." a bit hard to parse. In particular, the term "one distribution" is unclear. The next paragraph tries to explain why the decoder becomes almost deterministic; despite using words like "therefore" and "as a consequence," I found the explanations insufficient.

The term "overall encoding distribution" is unspecified in the paper,  and I am skeptical of the use of "true decoder likelihood." Maybe I am missing something obvious, but why does the expression for "true decoder likelihood" involve the variational approximation? Is it because we work under the assumption that the KL divergence is negligible?

Also, I am unsure what the objective is when you have "removed" the ELBO (see the first line, last paragraph, section 2.2.).



#### Minor bits:
"... we do not have *an unknown ..." --using an in place of a.

### Updates after the rebuttal

I find the revised manuscript to be clear and more transparent. After reading the reviews, the response, and the extended appendix, I am increasing my score and vote to accept this paper.

---

> ### Author Response · Authors · 2020-11-18
> **Response to Review3**
>
> Thank you for your detailed review and valuable feedback. Please find our answers below.
>
> **In the first paragraph, I find the last line: "Specifically, as normalizing .." a bit hard to parse. In particular, the term "one distribution" is unclear.**
>
> Normalizing Flows usually work on continuous data due to the rule of change of variable which is naturally defined for continuous variables. Therefore, we first need to cast the categorical data into the continuous space before applying a flow on them. However, the categories cannot be assigned a single value, as in this case, the flow would learn to model delta peaks on these specific values, and we would obtain arbitrarily high likelihoods. Thus, categories need to mapped to a distribution in continuous space so that the flow needs to model a volume for each categorical value instead of single numbers. The term "one distribution" refers to this encoding distribution we learn for the categorical input $x$. While we model an encoding distribution, we still train the models by sampling from the distribution. Hence, the inputs to the flow are still single numbers, but as those are sampled from the encoding distribution, the flow cannot place all probability mass on a single value. We clarify this sentence in the first paragraph of Section 2.2.
>
> **The next paragraph tries to explain why the decoder becomes almost deterministic; despite using words like "therefore" and "as a consequence," I found the explanations insufficient.**
>
> If the distributions of different categories overlap in the latent space, the decoder cannot be deterministic. If distributions overlap, it also means that the normalizing flow prior $p(z)$ cannot distinguish between all categories based on the continuous latent space $z$. However, this is crucial to allow the prior to model any possible dependency between categorical variables $x$. As the decoder is factorized over latent variables, it cannot model any dependencies between categorical variables. Meanwhile, the prior contains the main modeling complexity of the framework. Thus, we want the encoding to be optimized for it to model any necessary dependency between variables. Note that the normalizing flow in the prior is expressive enough to model any stochasticity that the decoder could if it would be necessary; for instance if there is an overlap of categories on a factorized level. In the Categorical Normalizing Flow framework, we have experienced that the model automatically catches up on this dynamic. The encoder separates the distributions of different categories in order to use the full expressiveness of the prior. The decoder, therefore, becomes almost deterministic, also because this minimizes the negative log-likelihood objective with respect to the decoder's parameters.
>
> As an example, we can assume that we have two categories $A$ and $B$, and each data point in the dataset has two elements, i.e. $\\mathbf{x}=\\{x_1,x_2\\}$. We choose the example dataset to be $\mathcal{D}=[\\{A,A\\}, \\{B,B\\}]$. Thus, the model must learn to model dependencies between $x_1$ and $x_2$. If the encoding distributions of $A$ and $B$ overlap identically, we cannot learn any dependency between $x_1$ and $x_2$. This is because the prior cannot distinguish between $A$ and $B$ in latent space, and the decoder cannot model any dependencies between $x_1$ and $x_2$. As long as the distributions of $A$ and $B$ noticeably overlap, there remains noise on the encoding. This noise propagates to the decoder making decoding stochastic. Due to this uncertainty, the prior has to assign a non-zero likelihood to the data points $\\{A,B\\}$ and $\\{B,A\\}$.
> In the end, the only way to model this dataset is by clearly separating the encoding distributions for $A$ and $B$, and therefore obtaining almost deterministic decoding.
>
> We clarified this explanation in Section 2.2 of the updated paper.
>
> **The term "overall encoding distribution" is unspecified in the paper**
>
> By "overall encoding distribution" we refer to $q(z_i)$, i.e. the combined encoding distribution of all category-dependent distributions: $q(z_i)=\sum_{\hat{x}} p(\hat{x})q(z_i|\hat{x})$. In the particular case in the paper, this is a mixture of $K$ logistic distributions, where each $q(z_i|\hat{x})$ represents one logistic mode. We have rephrased the corresponding sentence in the paper.

---

> > ### Author Response · Authors · 2020-11-18
> > **Response to Review3 (continuing)**
> >
> > **I am skeptical of the use of "true decoder likelihood." Maybe I am missing something obvious, but why does the expression for "true decoder likelihood" involve the variational approximation? Is it because we work under the assumption that the KL divergence is negligible?**
> >
> > The "true decoder likelihood" was intended to refer to the decoder that is reflecting the factorized posterior of the encoding distribution, i.e. $q(x_i|z_i)$. We called it "true" because the decoder is not learned by yet another network, but instead calculated based on the encoding distribution. The variational approximation still remains with this decoder because the variational gap is based on the joint latent space $z$, while the encoder and decoder work on the factorized space $z_1,...,z_S$.
> >
> > **Also, I am unsure what the objective is when you have "removed" the ELBO (see the first line, last paragraph, section 2.2.).**
> >
> > The K-logistic mixture encoding allows us to formulate the decoder $p(x_i|z_i)$ as a function of the encoder $q(z_i|x_i)$: $p(x_i|z_i)=\frac{\tilde{p}(x_i)q(z_i|x_i)}{\sum_{\hat{x}}\tilde{p}(\hat{x})q(z_i|\hat{x})}$. What we mean, therefore, is that there is no mismatch between the encoder's reconstruction likelihood, i.e. $q(x_i|z_i)$, and the decoder $p(x_i|z_i)$. That said, indeed there still exists a KL divergence between the joint approximate and true posterior, i.e. $D_{KL}\left(q(z|x)||p(z|x)\right)\geq0$, as the true posterior $p(z|x)$ cannot be factorized due to the joint prior $p(z)$. The true posterior can however be formulated as follows:
> > $$
> >     \begin{split}
> >         p(z|x) & = \frac{p(x|z)p(z)}{p(x)} \\
> >         & = \left(\prod_i \frac{q(z_i|x_i)p(x_i)}{\sum_{\hat{x}} q(z_i|\hat{x})p(\hat{x})}\right)\frac{p(z)}{p(x)}\\
> >         & = q(z|x) \cdot \frac{p(z)}{p(x)} \cdot \prod_i\frac{p(x_i)}{q(z_i)} \\
> >     \end{split}
> > $$
> > So, what we mean is that there is a KL term, which relates to the above formulation. Specifically, a change in $q(z|x)$ is directly propagated to the true posterior $p(z|x)$, which potentially simplifies the optimization of the model further. The KL divergence solely relies on the difference between the mismatch of joint prior and evidence, and not additionally on the encoder-decoder mismatch. Especially at the beginning of the training, we see in experiments much faster learning with the K-logistic mixture encoding because of the encoder and decoder share parameters.
> >
> > The objective of the mixture encoding is thereby the following ELBO:
> > $$
> > E_{x\sim P_{\text{data}}}\left[\log P_{\text{model}}(x)\right] \geq E_{x\sim P_{\text{data}}}E_{z\sim q(\cdot|x)}\left[\log \left(p_{\text{model}}(z) \prod_{i=1}^{S} \frac{\tilde{p}(x_i)}{\sum_{\hat{x}}\tilde{p}(\hat{x})q(z_i|\hat{x})}\right)\right]
> > $$
> > We clarify these parts in Section 2.2 of the paper, and added a more detailed section in Appendix A.1.1.
> >
> > **We do not have "an" unknown KL divergence adding the details about the metric (and the direction of improvement) in Tables 3 and 4.**
> >
> > In Table 3 (in the updated paper Figure 4), we compared LatentNF and the proposed CNF purely on their encoding. The reconstruction error is the decoder bits per dimension score, meaning the uncertainty in decoding. A low reconstruction error points to an almost deterministic decoder. Overall, the results show that the direction of improvement is given by the encoding of the CNF, and hence towards simpler encoding functions.

---

### Official Review · AnonReviewer1 · 2020-10-28
**ICLR 2021 Conference Paper1592 AnonReviewer1**

**Rating:** 6
**Confidence:** 3

**Review:**

# Summary #

This paper proposes a new normalizing flow model for categorial data, where the typical dequantization is not applicable.

We assume a categorical sample x has S variables, and each attribute x_i is a categorical variable.
We want to model the probability mass of the S variable categorical data, and devise an invertible map that can convert x into the continuous latent variable z.
For simplicity, we assume that each attribute x_i has its own latent continuous probability distribution p(z_i).
We expect an encoder, q(z_i|x_i), map categorical x into a continuous space where all categories are well partitioned. For that purpose, the paper proposes to formulate q(z_i|x_i) by a mixture of logistic distributions.

A graph generative flow model is proposed as an application.
Existing flow models for graphs do not handle the categorical data in proper manners and are permutation-dependent.
The proposed categorical flow can develop a permutaion-invariant graph generative flow model.

The proposed model performs better than the existing graph flow models in the molecular graph generation experiments. Typical invalid generation examples include isolated nodes. If we only focus on the largest sub-graphs, the proposed model can almost perfect graph generations.
The permutation-invariant nature of the proposed model results in a stable performance on the graph coloring problem, while the baseline RNN models are deeply affected by the order choices.

# Comment

I found the mixture of logistic regression is a good idea. Figure 2 and 3 in appendix indicate that this formulation can pratitioin the latent space into categories.

I have a few questions to confirm my understanding of the paper.

Q1. The proposed categorical normalizing flow with K-logistic mixture provides an approximated invertible map for the true distribution of the categorical samples x. Is this correct? Namely, there is a non-zero KL divergence between the evidence P(X) and the marginalized ``likelihood'' q(X)??

Q2. The paper says "we do not have a unknown KL divergence between approximate and true posterior constituting an ELBO". Does this mean we can compute the KL(q||p) analytically for the categorical normalizing flow?

Concerning the Q2. it is better if the final objective function to maximize/minimize, and the actual procedure for model training is clearly written in the main manuscript or the appendix.

Concerning the molecular graph generation experiments, I'm interested in how the latent representations of the graphs are distributed in the space of Z. It is preferable if the paper can provide a visualization of the latent space for the actual molecular graph generation experiments, not the simulated ones of Figure 2 and 3.

Presentaions of the experimental results in the main manuscript totally rely on the tables. However, current tables are not much effective to tell the significance of the proposed method.
Please consider visual presentations: we may use plots or bar graphs to compare several methods for example if the detailed numbers are not important.
The actual numbers can be moved to the appendix.

# Evaluation points

(+) A new approach to apply the normalizing flow.

(+) Truly permutation-invariant NF for graph generation is great.

(-) insufficient explanations for the optimization procedure

(-) more visual results may improve impressions of the manuscript (especially for non-expert readers)

---

> ### Author Response · Authors · 2020-11-18
> **Response to Review1**
>
> Thank you for your detailed review and valuable feedback. Please find our answers below.
>
> **Q1. The proposed categorical normalizing flow with K-logistic mixture provides an approximated invertible map for the true distribution of the categorical samples x. Is this correct? Namely, there is a non-zero KL divergence between the evidence P(X) and the marginalized ``likelihood'' q(X)?**
>
> Yes, it is correct that there is a non-zero KL divergence between the evidence $P(X)$ and the marginalized likelihood $q(X)$. The variational gap is caused by the Jensen inequality ($\log\mathbb{E}[...]\geq\mathbb{E}[\log...]$), and is in this aspect similar as variational dequantization for image modeling.
>
> **Q2. The paper says "we do not have a unknown KL divergence between approximate and true posterior constituting an ELBO". Does this mean we can compute the $KL(q||p)$ analytically for the categorical normalizing flow?**
>
> The K-logistic mixture encoding allows us to formulate the decoder $p(x_i|z_i)$ as a function of the encoder $q(z_i|x_i)$: $p(x_i|z_i)=\frac{\tilde{p}(x_i)q(z_i|x_i)}{\sum_{\hat{x}}\tilde{p}(\hat{x})q(z_i|x)}$. What we mean, therefore, is that there is no mismatch between the encoder's reconstruction likelihood, i.e. $q(x_i|z_i)$, and the decoder $p(x_i|z_i)$. That said, indeed there still exists a KL divergence between the joint approximate and true posterior, i.e. $D_{KL}\left(q(z|x)||p(z|x)\right)\geq0$, as the true posterior $p(z|x)$ cannot be factorized due to the joint prior $p(z)$. The true posterior can however be formulated as follows:
> $$
>     \begin{split}
>         p(z|x) & = \frac{p(x|z)p(z)}{p(x)} \\
>         & = \left(\prod_i \frac{q(z_i|x_i)p(x_i)}{\sum_{\hat{x}} q(z_i|\hat{x})p(\hat{x})}\right)\frac{p(z)}{p(x)}\\
>         & = q(z|x) \cdot \frac{p(z)}{p(x)} \cdot \prod_i\frac{p(x_i)}{q(z_i)} \\
>     \end{split}
> $$
> So, what we mean is that there is a KL term, which relates to the above formulation. Specifically, a change in $q(z|x)$ is directly propagated to the true posterior $p(z|x)$, which potentially simplifies the optimization of the model further. The KL divergence solely relies on the difference between the mismatch of joint prior and evidence, and not additionally on the encoder-decoder mismatch. Especially at the beginning of the training, we see in experiments much faster learning with the K-logistic mixture encoding because of the encoder and decoder share parameters.
>
> We clarify these parts in Section 2.2 of the paper and added a more detailed section in Appendix A.1.1.
>
> **Concerning the Q2. it is better if the final objective function to maximize/minimize, and the actual procedure for model training is clearly written in the main manuscript or the appendix**
>
> Thank you, we add the final lower bound objective for the mixture encoding in Equation (3) in the updated paper:
> $$
> E_{x\sim P_{\text{data}}}\left[\log P_{\text{model}}(x)\right] \geq E_{x\sim P_{\text{data}}}E_{z\sim q(\cdot|x)}\left[\log \left(p_{\text{model}}(z) \prod_{i=1}^{S} \frac{\tilde{p}(x_i)}{\sum_{\hat{x}}\tilde{p}(\hat{x})q(z_i|\hat{x})}\right)\right]
> $$
> During training, we therefore minimize the negative log-likelihood loss of the lower bound:
> $$
> \mathcal{L}=-\log \left(p_{\text{model}}(z) \prod_{i=1}^{S} \frac{\tilde{p}(x_i)}{\sum_{\hat{x}}\tilde{p}(\hat{x})q(z_i|\hat{x})}\right)
> $$
> We also add an algorithmic overview on how to train and test a Categorical Normalizing Flow in Appendix A.1.2.
>
> **Concerning the molecular graph generation experiments, I'm interested in how the latent representations of the graphs are distributed in the space of Z. It is preferable if the paper can provide a visualization of the latent space for the actual molecular graph generation experiments, not the simulated ones of Figure 2 and 3.**
>
> We add visualizations for the encoding latent space of the edge attributes in molecules. We focus on the edge attributes because the attribute categories are only three ("single bond", "double bond", and "triple bond"). Thus, a two-dimensional latent vector $z_i$ was used in our experiments, which can be plotted in a 2D space. We include the visualization of the encoding latent distribution of the edge attributes to Appendix A.1.3.
> The visualization is similar to the simulated cases in the previous Figures 2 and 3 in the sense that the three categories have been separated across the latent space. One observation, consistent over multiple runs, is that due to the high category imbalance (75\% of the edges being single bonds, and only 0.2\% triple bonds), most of the latent space has a high decoding probability for the first category.

---

> > ### Author Response · Authors · 2020-11-18
> > **Response to Review1 (continuing)**
> >
> > **Presentations of the experimental results in the main manuscript totally rely on the tables. However, current tables are not much effective to tell the significance of the proposed method. Please consider visual presentations: we may use plots or bar graphs to compare several methods for example if the detailed numbers are not important. The actual numbers can be moved to the appendix.**
> >
> > We thank the reviewer for the suggestion and have replaced the results of the language modeling experiment with a bar plot (Figure 4), as the differences between models are indeed more important than the exact numbers. The table with the detailed numbers has been moved into Appendix D.4.

---

### Official Review · AnonReviewer2 · 2020-10-28
**categorical flows**

**Rating:** 7
**Confidence:** 4

**Review:**

The paper investigates normalizing flows for the general case of categorical data. The authors propose continuous encodings in which different categories correspond to unique volumes in a continuous latent space.

Using the proposed Categorical Normalizing Flows (CNF) the authors present GraphCNF, a permutation-invariant normalizing flow on graph generation.

In particular the paper presents a new approach for normalizing flow on categorical data. Then a three step generation approach is presented for graph generation with CNF.

The paper is well written and the proposed solution is clearly presented in sections 2 and 3.

The first results on molecule generation are very good when compared to those obtained with other approaches. The second experiment on graph coloring shows the validity of the proposed CNF even on an NP-hard optimization problem. Even the results on language modeling are encouraging. A final experiment shows how CNF can model discrete distributions precisely.

An experiment could be done on real-world datasets usually used for density estimation in order to assess the validity of CNF in therms of likelihood.

---

> ### Author Response · Authors · 2020-11-18
> **Response to Review2**
>
> Thank you for your kind words and suggestions. Please find our answer below.
>
> **An experiment could be done on real-world datasets usually used for density estimation in order to assess the validity of CNF in terms of likelihood.**
>
> Based on your suggestion, we have performed an experiment on a credit-card risk record dataset [1]. The data contains different categorical attributes regarding potential credit risk, and we have used the following 9 attributes: "checking_status", "credit_history", "savings_status", "employment", "housing", "job", "own_telephone", "foreign_worker" and "credit_risk". The task is to model the joint density function of those 9 attributes over a dataset of 1000 entries. Our results are:
>
> | **Model**                            | **Likelihood in bpd**                                              |
> |--------------------------------------|-------------------------------------------------------------------|
> | Variational Dequantization | $1.95$ $\pm0.04$                                                    |
> | Latent NF                               | $1.36$ $\pm0.03$ ($0.01$)                                       |
> | Categorical NF                      | $1.37$ $\pm0.03$ ($0.00$)                                       |
>
> The results have been averaged over three seeds, and the reconstruction error is shown in brackets where applicable. Both the LatentNF and CNF perform equally well, given the small size of the dataset and number of categories. However, we experienced a relatively high standard deviation due to the small size of the training and test set.
> The experiment and details about it have been added to Appendix D.5.
>
>
> [1] Dua, D. and Graff, C. (2019). UCI Machine Learning Repository [http://archive.ics.uci.edu/ml]. Irvine, CA: University of California, School of Information and Computer Science.

---

### Official Review · AnonReviewer4 · 2020-11-03
**Proposes normalizing flow algorithms for modeling discrete distribution**

**Rating:** 7
**Confidence:** 4

**Review:**

Summary:
The paper considers the problem of modeling discrete distributions with normalizing flows. Authors propose a novel framework “Categorical Normalizing Flows”, i.e CNF. By jointly modeling a mapping to continuous latent space, and the likelihood of flows CNF solves some of the bottlenecks in current algorithms. With experiments on some synthetic domains, and benchmarking tasks like Zinc250, the authors empirically demonstrate that CNF-based algorithms perform competitively and often improve significantly on related approaches like Latent NF, discrete flows.

Reasons for the score:
I vote for accepting the paper, with minor improvements. The problem is well motivated, and the proposed algorithm improves significantly on previous approaches. I would encourage the authors to include generated samples/visualizations which may reflect some pathological failure modes of the algorithm. This would help with the readability of the paper, and improve understanding.

Strengths:
+ The CNF framework helps scaling normalizing-flow based approaches, and improves the stability and performance on standard benchmarks.
+ GraphCNF outperforms baselines like GraphAF/GraphNVP on Zinc250 molecule generation.
+ On language modelling tasks, CNF improves on Latent NFs and works competitively as an LSTM baseline.

Weaknesses:
- Plots such as comparing the training loss of different approaches would help understanding the stability of the learning algorithm.
- Adding some visualizations/samples from the learned models would help with clarity of section5.

---

> ### Author Response · Authors · 2020-11-18
> **Response to Review4**
>
> Thank you for your encouragement and valuable feedback. Please find our answers below.
>
> **Plots such as comparing the training loss of different approaches would help understanding the stability of the learning algorithm.**
>
> We have added the loss plots of LatentNF and the proposed CNF on the text8 and Wikitext103 dataset to Appendix A.4.2 (Figure 9 and 10), and referenced it in the discussion of the main paper. The plots show the instability of LatentNF during training due to the high loss scheduling, which is not the case for CNF.
>
> **Adding some visualizations/samples from the learned models would help with clarity of section 5. I would encourage the authors to include generated samples/visualizations which may reflect some pathological failure modes of the algorithm. This would help with the readability of the paper, and improve understanding.**
>
> In the updated version of the paper, we have added to Section 5.1 (Molecule Generation, Figure 2) a visualization of a molecule of the dataset, a generated molecule, and an example for the failure case of generating multiple valid sub-graphs. In addition, Appendix C contains more generated molecules (Figure 11) and additional failure case examples beyond the sub-graphs (Figure 12). The failure cases mainly rely on single bonds that have been incorrectly predicted. For example, the model generates a double bond for an edge in the graph which results in a carbon atom having 5 connections although only allowing 4. If the model would have predicted a single bond, the molecule would have been valid. Besides, in Section 5.2 (Graph Coloring, Figure 3), we have added a small example of a graph in the dataset, with the corresponding coloring having been generated by a GraphCNF.
> We hope that this adds a visual explanation of the task.

---

### Author Response · Authors · 2020-11-18
**General response**

We would like to thank all reviewers for their valuable feedback. We have updated our manuscript with the following changes:
* Parts of Section 2.2 have been rephrased to improve clarity on the method/framework description
* Section 5 has been improved by adding more visual aspects. An example of molecule generation and graph coloring is added, as well as the results of the language modeling being replaced by a bar plot.
* The appendix has been extended based on the reviewer's questions with more details on the mixture encoding of CNF (A.1), the training behavior of CNF and Latent NF (A.4.2), more failure cases on molecule generation (Figure 12), and an additional experiment on real-world data (D.5).

We have added the appendix to the main PDF submission to allow easier access to it.

---

### Decision · Program_Chairs · 2021-01-07
**Final Decision**

**Decision:**

Accept (Poster)

**Comment:**

Well-written paper that proposes a flow-based model for categorical data, and applies it to graph generation with good results. Extending flow-models to handle types of data that are not continuous is a useful contribution, and graph generation is an important application. Overall, the reviewers were positive about the paper, and only few negative points were raised, so I'm happy to recommend acceptance.